# Seed dispersal by Martu peoples promotes the distribution of native plants in arid Australia

Rebecca Bliege Bird [1] ✉, Douglas W. Bird[1], Christopher T. Martine[2], Chloe McGuire[3], Leanne Greenwood[4], Desmond Taylor[5], Tanisha M. Williams[6] & Peter M. Veth [7]

Commensal relationships between wild plants and their dispersers play a key ecological and evolutionary role in community structure and function. While non-human dispersers are often considered critical to plant recruitment, human dispersers have received much less attention, especially when it comes to non-domesticated plants. Australia, as a continent historically characterized by economies reliant on non-domesticated plants, is thus a key system for exploring the ecological role of people as seed dispersers in the absence of agriculture. Here, we utilize a controlled observation research design, employing ecological surveys and ethnographic observations to examine how seed dispersal and landscape burning by Martu Aboriginal people affects the distribution of three preferred plants and one (edible, but non-preferred) control species. Using an information theoretic approach, we find that the three preferred plants show evidence of human dispersal, with the strongest evidence supporting anthropogenic dispersal for the wild bush tomato, *Solanum diversiflorum*.

The modes by which people modify "wild" plant distributions outside of the domestication process are widely debated and poorly understood[1], with most work focusing on the contemporary role of human mobility in passively spreading the seeds of unwanted weeds[2,3]. Recent studies have, however, provided intriguing deep-time evidence that sedentary, agriculturally-based indigenous communities may have been significant agents in the dispersal of some useful, non-domesticated plants, which are at higher density near old settlements and archeological sites or are hyperdominant—more extensively distributed geographically than expected[4–11]. Despite this work, very few studies have explored whether "hunter-gatherers" (people with highly mobile economies who do not make gardens) can also affect plant distributions through mechanisms outside of agriculture, such as seed dispersal or landscape disturbance.

Given the deep prehistory of hunting and gathering, and the profound evolutionary and ecological consequences of seed dispersal[12], investigating how hunter-gatherer seed dispersal might shape plant communities could significantly improve our understanding of the transition from foraging to farming[13], help design better biodiversity conservation policy[14], and eludicate the processes that lead to the formation of Indigenous cultural landscapes[15].

One of the most significant regions for exploring nonagricultural interactions with wild plants is Australia, where pre-contact economies across the continent were structured by foraging strategies accompanied by high residential mobility, no reliance on domesticated plants until the post-colonial period, and the extensive use of landscape fire. Aboriginal populations are thought to have influenced the distribution

[1]The Pennsylvania State University, Department of Anthropology, University Park, PA 16801, USA. [2]Department of Biology, Bucknell University, 1 Dent Drive, Lewisburg, PA 17837, USA. [3]Far Western Anthropological Research Group, 2727 Del Rio Pl, Davis, CA 95618, USA. [4]Dja Dja Wurrung Clans Aboriginal Corporation, Bendigo, Australia. [5]Martu Elder, Kulyakartu Aboriginal Corporation, 76 Wittenoom St, East Perth, WA 6004, Australia. [6]Department of Plant Biology, University of Georgia, 2502 Miller Plant Sciences, Athens, GA 30602-7271, USA. [7]The University of Western Australia, School of Social Sciences and ARC Centre of Excellence for Indigenous and Environmental Histories and Futures, Crawley, WA, Australia. ✉e-mail: rub33@psu.edu

of at least 50 plants[16] of economic, medicinal, or ritual importance through modifications to the fire regime[17–19], bioturbation[20], and seed scattering[21,22]. Studies of landscape-level genomic diversity in Australia have provided indirect evidence for a human dispersal role for the Australian baobab[23] (*Adansonia gregorii*), bunya pine (*Araucaria bidwillii*)[24], and black bean tree[25] (*Castanospermum australe*). Ethnographic observations of bush tomato (*Solanum* spp.) foraging and consumption have provided additional anecdotal evidence for seed dispersal[26–29]: Kimber[26] noted that Aboriginal foragers often discarded seed inadvertently around camps and sometimes carried *Solanum* fruit considerable distances, a practice which he speculated "must have occasionally aided dispersal". In the Central desert, O'Connell and Latz[27] noted that one bush tomato species, *S. chippendalei*, had been introduced and become abundant in areas north and west of its natural range by Aboriginal people transporting fruit and seed to remote settlements by vehicle. They also noted that there were many fruits with inedible seeds (such as *Capparis* spp., *Canthium* spp., and *Carissa* spp.) that were stored underground in caches in large quantities, again suggesting a possible dispersal role.

Here, we build on this work by employing a hypothetico-deductive methodology with a controlled comparison research design to test potential causal mechanisms for anthropogenic influence on the dispersal of edible plants on a Western Australian desert landscape with a long history (ca 50k years) and continued practice of highly mobile hunting and gathering[30]. Using ground-based ecological transect surveys in conjunction with proxies for seed dispersal and anthropogenic disturbance in both past and present, we ask how a history of Aboriginal landscape use by Martu peoples has affected the distribution (presence and abundance) of four edible plants: two bush tomatoes (*Solanum diversiflorum* and *S. centrale*), a seed grass (*Eragrostis spp*), and a small forb (*Scaevola parvifolia*). We predict that *S. diversiflorum* and *Eragrostis* will show more signs of anthropogenic dispersal in both past and present as *S. div* seeds are inedible and need to be removed during processing, and *Eragrostis* seeds are small and easily lost during the winnowing process (see Methods). We predict that *S. centrale* dispersal potential is weaker due to the edible nature of its seeds, and that *Scaevola* will not show signs of dispersal due to its low preference rank. All four may be affected by Martu disturbance in the form of landscape fire. This analysis is part of a long-term collaborative ethnographic, ecological, and ethnoarchaeological project with Martu communities located in the heart of their homelands in Western Australia. Martu Native Title encompasses more than 150,000 sq km of the Great and Little Sandy Deserts ecoregions[31], and our study area comprises a subset region of about 42,000 sq km (Fig. 1G). See Methods for more ethnographic details. Our results show that Martu seed dispersal processes in past and present have substantially affected the distribution of *S. diversiflorum* and *Eragrostis* grasses, while the presence of *S. centrale* depends on legacies of anthropogenic landscape fire in the form of increased fire frequency. There are no dispersal or fire effects on the control species, the little-used forb *Scaevola parvifolia*.

## Results

Data on plant presence and abundance comes from ten corridor transect surveys of plant presence and abundance conducted by one of the authors (RBB) throughout the Martu Native Title region[32], along with 12 quadrat surveys of observed *S. diversiflorum* dispersal sites. Our predictor variables are derived from remotely sensed environmental and soil rasters, fire history mapping, and contemporary ethnographic and ecological observations made over 25-year period with remote Martu communities. We first provide some qualitative and quantitative evidence for the potential seed dispersal derived from our ethnographic and ecological observations. We then use a model selection approach to build a "global" model predicting the presence of each species with the same set of dispersal proxies (land use) and ecosystem engineering proxies (fire predictors). Next, we find the best set of

presence predictors for each species including soil and moisture covariates. Finally, we find the best set of predictors for the abundance of each species when present.

## Harvesting and seed dispersal: ethnographic observations

Many plants are recognized by Aboriginal people as being dispersed through the activities of humans, animals, and ancestral beings[28]. In our many years of dialog with Martu while foraging, adults have often acknowledged their role in the propagation of useful plants, which has also been reported by others[22]. People offer various explanations for why some plants grow so densely in some places and not others by noting that this was where people used to camp and clean seed, or that a particular locale had been popular for fruit picking "for a long time". Continued use of the region is thought to have supported the growth of more plants. Alternatively, Martu have noted that fire keeps some plants coming back, and that a fire burned in winter produces more fruit. Many describe how as youngsters they were told to clean bush tomatoes within the same place (patch), so that the plants and their fruit/seeds would always be there. This does not translate to management of plants, per se, in the way that farmers might: Martu say they are acquiring and sharing resources with others as a meaningful means of building relational and ritual wealth[32–34]. In so doing, this process of making food good for consumption creates and shapes landscapes, which are both the source of food and socio-religious capital. It is this precise knowledge that has ritual significance: the knowledge that Martu belong to the country and to the plants and animals that live within it, and they have both a spiritual *and* material role to play in making that country productive.

Our participant observation of food harvesters revealed a high potential for dispersal for two of our four species. Over 385 foraging days from a total of 800 days in residence, we quantitatively recorded *S. diversiflorum* harvesting on 11 days in 5 different patches (51.22 person-h for 143.7 kg of whole fruit). *Solanum* fruit harvesting is a high-ranked foraging activity: the caloric value of *S. diversiflorum* is 920 kcal/kg[35] with 84% of the fruit edible, producing an in-patch foraging return of 2168 kcal/h/forager (total edible yield harvested by a single forager on a given bout of collecting and processing within a given patch), which is one of the most productive of all resources in the region[36] and saliently, more so than most hunting activities.

On foraging trips for S. *diversiflorum* fruit, a small group of harvesters traveled by vehicle to known dense patches (Fig. 1F), foraging on foot and consuming fruit within the patch. During these bouts, we observed substantial in-patch seed dispersal, as harvesters cleaned fruit for consumption while walking between clusters of plants within the patch (Fig. 1A). On some harvesting trips, fruit was loaded into vehicles and taken to the community for processing at home; seed dispersal often occurred en-route as foragers ate fruit and discarded seed out of the windows of the vehicle while traveling. On others, foragers transported harvests to a temporary dinner camp, where additional processing resulted in heavy seed dispersal around a central hearth (Fig. 1E). Fruit was often transported back to communities by 4WD vehicle, where additional processing caused seed to be discarded around residential areas. Contemporary seed transport happened most often along vehicle tracks, within communities, and at foraging camps within the daily range of Martu Aboriginal communities. In maintaining their especially high residential mobility, Martu will also sometimes transport a load (up to 10 kg) of unprocessed *S. diversiflorum* to share with kin in other communities, occasionally even beyond their Native Title boundaries. While we did not observe any deliberate dispersal of seed with the intention of creating a garden patch, we note that Fiona Walsh[28], observing Martu foraging in the mid-1990s, did find some deliberate dispersal of seed near settlements in order to bring patches closer to home.

To quantify the potential for anthropogenic dispersal of *S. diversiflorum*, we also conducted systematic surveys of 12 contemporary

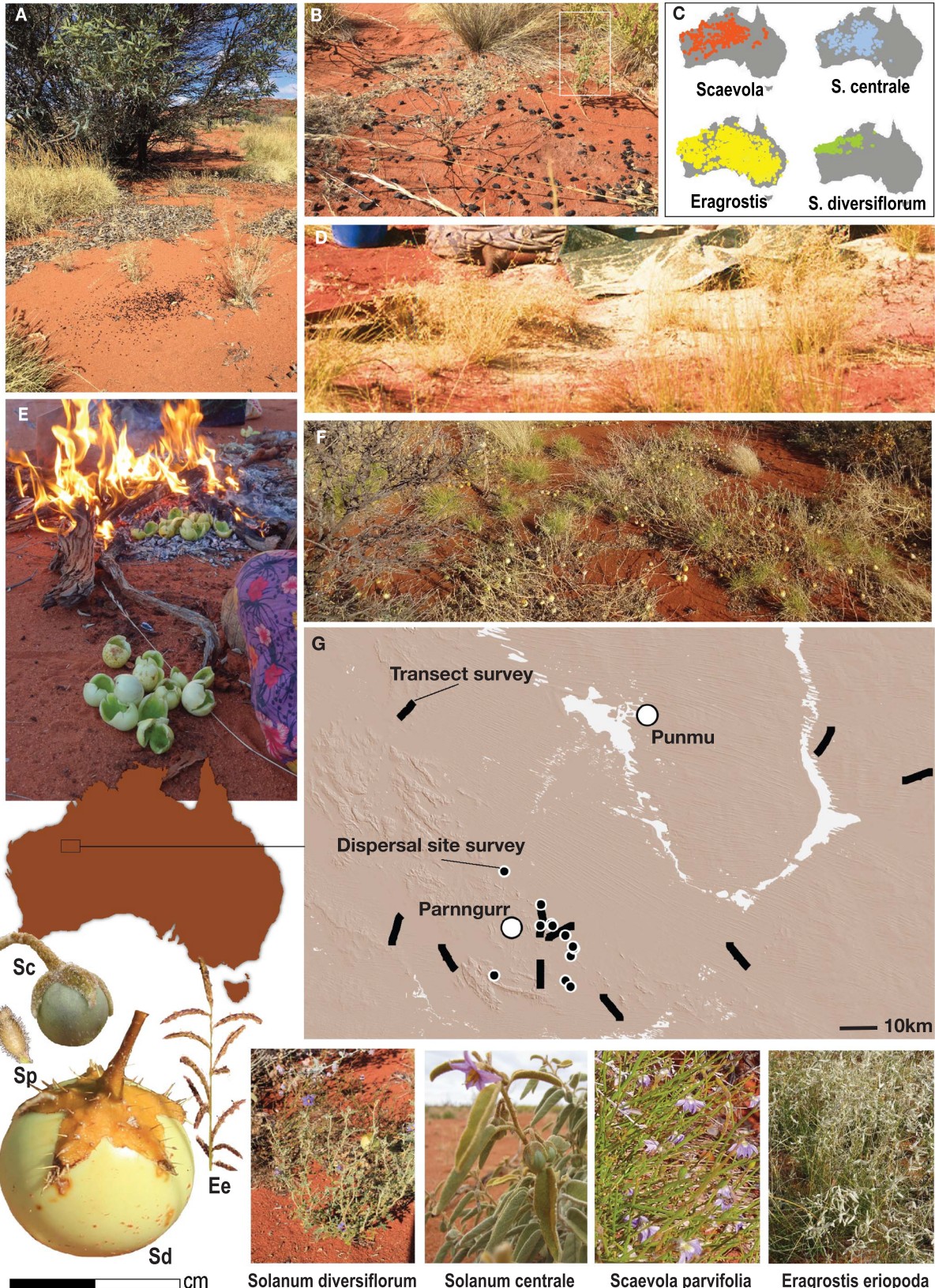

**Fig. 1 | Study location and study species. A** A scatter of black seeds in the foreground is evidence of a recent *S. diversiflorum* dispersal event. **B** A single *S. diversiflorum* plant (white box) at the edge of a 3-year old hearth where fruit was cleaned and seeds dispersed on a multi-day camp. **C** Occurrence records (derived from the Atlas of Living Australia) of all four species. **D** Eragrostis seed hulls scattered over the ground after threshing at camp. **E** Fruit being cleaned and cooked at the hearth after a long day of hunting. **F** A dense "garden" patch of *S. diversiflorum* with *E. eriopoda/setifolia* visible in the foreground. **G** Survey locations. The Transect survey area samples a 42,000 km² subset of the Martu native title region. Map services (DEM) and data available from U.S. Geological Survey. Lower left: A comparison of fruit size by species: Sd *S. diversiflorum*, Sc *S. centrale*, Ee *Eragrostis eriopoda*, Sp *Scaevola parvifolia*.

dinner camp locations (hearths used between 2002 and 2017 where bush foods are cooked and consumed away from the community) to test whether use of the site to process fruit increases plant density compared to dinner camps where fruit is not consumed. In 2018, we surveyed six former dinner camps where fruit was observed to be processed, and six where other bush foods were consumed. Our sample size was small, but plants were present at only two out of six locations where no collection had occurred, and at 83% of hearth locations (five out of six) with known consumption events. The presence of *S. diversiflorum* in the two most recent camps (2015 and 2017) was obvious, localized around the hearth area and other locations where seeds had been discarded in bulk. This is clearly observed in Fig. 1B, where plants are emerging in the ashes and charcoal of a 3-year old cooking hearth. Mean plant density at consumption sites was significantly higher at 1.902 stems/ha [95%CI 0.961, 1.424] compared to control sites at 0.251 stems/ha [95%CI −0.691, 1.931].

The contemporary dispersal potential for *S. centrale* was low even though the fruits were harvested whenever encountered. We observed *S. centrale* harvesters on 17 days in 12 different patches (120.25 person-h for 49.15 kg of whole fruit). *S. centrale* is also high ranked: 1360 kcal/kg with 100% edible, producing 556 kcal/h. Dispersal of seed via epizoochory was not obvious, and was limited to incidental losses of fruit from makeshift containers or drying platforms, as every tiny fruit was highly cherished by children.

We observed *Eragrostis* seed heads being harvested and processed occasionally, but never in any great quantities[37]. However, there was a moderately high potential for seed dispersal. Seed heads were collected into large dishes or bins and transported to camp for processing, which traditionally would involve threshing with a large rock on a hard flat termite bed[38]. We did observe a few threshing events that involved the use of a plastic tarpaulin to reduce seed loss while threshing. Following threshing, the loose seeds were swept up from the ground and winnowed to remove sand and other foreign material. Given the tiny size of the seeds, the amount of chaff flying around the processing area (see Fig. 1D), and the inefficiencies in winnowing, undoubtedly many seeds ended up lost in the soil nearby.

We did not observe any potential for direct epizoochoric dispersal of *Scaevola parvifolia*. On several foraging expeditions, people encountered patches, noted that the fruit was edible, mentioned it was an important food source for Australian bustard (*Ardeotis australis*) and burrowing bettong (*Bettongia leseur*, now locally extinct), but did not harvest it. In the early 1990s, when the outstation communities were just established, people collected a wider variety of plants than during our fieldwork (2000–present): even so, *Scaevola* was only observed to be collected once[28]. Several of us have tried the fruit, and it is sweet but very hard to collect as ripe fruit drops to the ground with a light touch, plants only produce a few fruits at a time, and the flesh gives an exceedingly small reward relative to the size of the seed.

## Statistical modeling of plant abundance and presence

In step one, employing a prior survey of plant presence and abundance conducted in 2003 by one of the authors (RBB), we operationalized and selected our predictor variables as noted in the Methods and constructed a set of explanatory hierarchical generalized linear models utilizing transect ID ($n = 10$) as random effect, and which included four dispersal proxies (archaeological/residential site distance, site type, contemporary land use intensity, nearest water permanence--the Dispersal Global Model) and three engineering proxies (time since fire (TSF) diversity, fire season, fire frequency--the Engineering Global Model) to see how well each set predicted plant presence for each species (Table 1). We refer to these as the global models, as they were applied to all four species regardless of how well the models fit. As all four species were generally absent in survey plots greater than 3.5 years after firing (Table 2), only data from plots that had been burned less then 3.5 years ago were used for modeling ($n = 2429$ plots of 300 m² each).

**Table 1 | Summary of hypothesis testing for the effects of anthropogenic seed dispersal on plant present/abundance**

| Species | Factors | Prediction |
|---|---|---|
| *Diversiflorum* | Past use: Site type | Minor sites higher |
| | Past use: Site distance | Strong negative effect |
| | Past use: Water permanence | Strong positive effect |
| | Present use: Fire density | Strong positive effect |
| | Fire: TSF diversity | Null: no effect |
| | Fire: Fire frequency | Null: no effect |
| | Fire: Fire season | Null: no effect |
| | Behavioral observations | High potential for seed dispersal Plants growing at dispersal sites |
| Centrale | Past use: Site type | No effect |
| | Past use: Site distance | Weak negative effect |
| | Past use: Water permanence | Weak positive effect |
| | Present use: Fire density | No effect |
| | Fire: TSF diversity | Null: no effect |
| | Fire: Fire frequency | Null: no effect |
| | Fire: Fire season | Null: no effect |
| | Behavioral observations | Potential for seed dispersal |
| *Eragrostis* | Past use: Site type | No effect |
| | Past use: Site distance | Weak negative effect |
| | Past use: Water permanence | Weak positive effect |
| | Present use: Fire density | No effect |
| | Fire: TSF diversity | Null: no effect |
| | Fire: Fire frequency | Null: no effect |
| | Fire: Fire season | Null: no effect |
| | Behavioral observations | Potential for seed dispersal |
| *Scaevola* | Past use: Site type | No effect |
| | Past use: Site distance | No effect |
| | Past use: Water permanence | No effect |
| | Present use: Fire density | No effect |
| | Fire: TSF diversity | Null: no effect |
| | Fire: Fire season | Null: no effect |
| | Fire: Fire frequency | Null: no effect |
| | Behavioral observations | Low potential for seed dispersal |

Predictions for each anthropogenic factor used in the statistical modeling and for the ethnographic observations of dispersal behavior for each species. Anthropogenic factors are proxies for dispersal behaviors (past and present Martu land use) and landscape engineering through fire. See Methods for more details on hypothesis development.

In the engineering global model, winter fires and TSF diversity strongly increased the presence of *S. diversiflorum*, while fire frequency increased S. *centrale* presence. Engineering effects on *Eragrostis* were mixed and subject to complex interaction effects that were difficult to tease out. Presence increased following more frequent fires, but only when these occurred in winter. Presence was less likely with higher TSF diversity. *Scaevola* presence was weakly increased following winter fires but did not respond to any other predictors. Pseudo-R² (Nakagawa) values suggest the fixed effects (excluding the random effect of transect location) of engineering explain the most variation in both Solanum species compared to the other edible species (Fig. 2, Table 3).

**Table 2 | Transect survey raw data**

|  | *Solanum diversiflorum* | *Solanum centrale* | *Scaevola parvifolia* | *Eragrostis setifolia* |
|---|---|---|---|---|
| *N* 30 m Plots | 3130 | 3130 | 3130 | 3130 |
| Plots present | 109 | 70 | 177 | 617 |
| Present by % total plots surveyed (*n*) |  |  |  |  |
| Time Since Fire: 0–1.75 years | 7.06% (67) | 4.11% (39) | 4.95% (47) | 31.82% (302) |
| 1.75–3.25 years | 2.57% (34) | 1.97% (26) | 8.54% (113) | 22.98% (304) |
| 3.25–5.75 years | 1.26% (8) | 0.79% (5) | 2.05% (13) | 1.26% (8) |
| 5.75+ years | 0.00%(0) | 0.00% (0) | 1.79% (4) | 1.34% (3) |

Percentage of plots where species was present by time since fire.

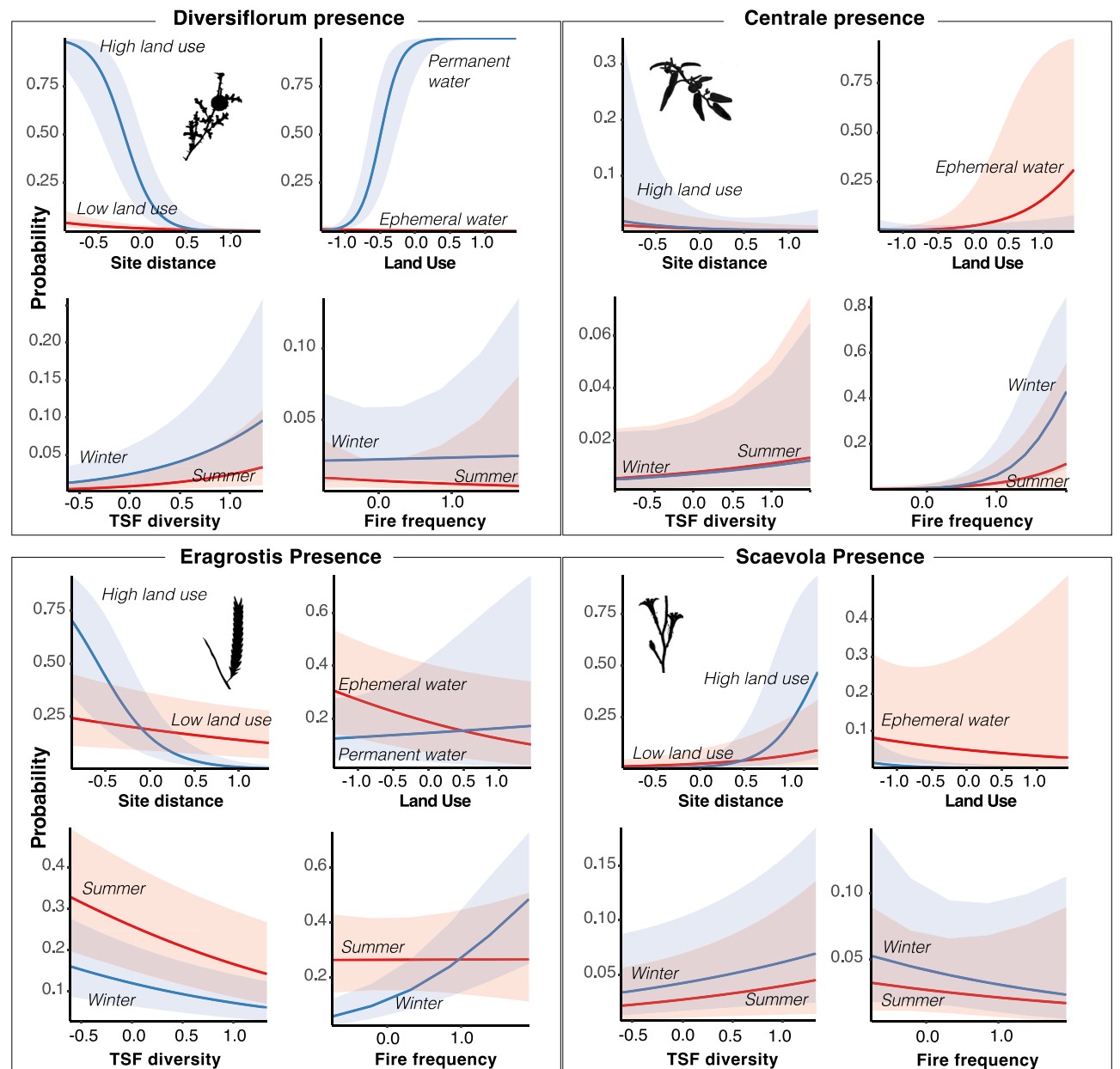

**Fig. 2 | Predicted probabilities of plant presence.** Model predicted values of plant presence in the global models (see values in Table 3) by distance to nearest residential/archeological site (at high and low land use), intensity of contemporary land use (at permanent and ephemeral water), time since fire diversity (by season of most recent fire), and fire frequency (by season of most recent fire). Predictor variables are standardized such that a value of 1 equals 2 standard deviations. Shaded error bands indicate 95% confidence intervals for the predicted value.

**Table 3 | Global engineering and dispersal model results**

|  | Diversiflorum | Centrale | Eragrostis | Scaevola |
|---|---|---|---|---|
| Engineering Model |  |  |  |  |
| Fire season: winter | 3.02 ** [1.48, 6.15] | 0.87 [0.39, 1.93] | 0.37 *** [0.28, 0.49] | 1.59 * [1.04, 2.44] |
| Fire Frequency | 0.69 [0.17, 2.74] | 4.06 ** [1.48, 11.16] | 1.04 [0.69, 1.56] | 0.80 [0.38, 1.67] |
| TSF diversity | 3.08 *** [1.75, 5.45] | 1.50 [0.69, 3.29] | 0.56 *** [0.42, 0.74] | 1.46 [0.91, 2.35] |
| TSFdiv × Fire freq | 0.58 [0.18, 1.83] | 0.47 [0.16, 1.38] | 1.57 [0.94, 2.61] | 1.43 [0.58, 3.56] |
| Season × Fire freq | 1.47 [0.33, 6.57] | 2.61 [0.79, 8.62] | 2.73 *** [1.56, 4.77] | 0.93 [0.35, 2.48] |
| $R^2$ Fixed effects | 11.44% | 12.07% | 7.19% | 1.98% |
| $R^2$ Fixed + Random | 41.12% | 56.94% | 30.52% | 40.17% |
| Dispersal Model |  |  |  |  |
| Site distance | 0.27 *** [0.15, 0.49] | 0.38 * [0.16, 0.91] | 0.69 * [0.49, 0.98] | 3.49 *** [1.94, 6.29] |
| Land use | 17.01 *** [6.50, 44.50] | 1.20 [0.36, 4.02] | 0.77 [0.50, 1.17] | 0.46 [0.20, 1.07] |
| Major site | 0.08 * [0.01, 0.58] | 1.04 [0.06, 18.04] | 1.80 * [1.05, 3.07] | 0.19 * [0.05, 0.71] |
| $H_2O$ Perm | 3.84 * [1.04, 14.24] | 1.34 [0.22, 8.27] | 0.40 * [0.19, 0.87] | 0.09 ** [0.02, 0.43] |
| Land use × $H_2O$ Perm | 2836.02 *** [314.78, 25551.36] | 0.03 * [0.00, 1.00] | 1.84 [0.58, 5.89] | 0.21 [0.03, 1.54] |
| Site dist × Land Use | 0.01 *** [0.00, 0.09] | 0.62 [0.04, 8.69] | 0.08 *** [0.03, 0.20] | 12.67 ** [2.30, 69.84] |
| $R^2$ Fixed effects | 44.88% | 14.88% | 8.75% | 19.05% |
| $R^2$ Fixed + Random | 55.50% | 68.10% | 43.54% | 68.79% |

Model covariates (GLM, binomial, $n$ = 10 random effect levels, $n$ = 2924 observations) as the standardized odds ratio for the best presence models (±95% CI in square brackets). The odds ratios reported here give the relative change in the likelihood of presence for a two-standard deviation increase in the predictor variable. Odds ratios higher than 1 indicate the predictor variable increases the odds of presence; those less than 1 indicate a negative relationship (the predictor variable decreases the likelihood of presence). Interaction (crossed) terms in the model are represented by "X". Caution must be used in interpreting the coefficients from interaction terms. $R^2$ terms are reported as Nakagawa's $R^2$ and are suggestive of the proportion of variation explained by the fixed terms, vs the fixed terms plus the random effect of Transect. Predicted values for some variables are shown in Fig. 2. Asterisks indicate significant $p$ values (* = <0.05, ** = <0.01, *** = <0.001).

In the dispersal global model, *S. diversiflorum* was strongly predicted to be present close to sites and presence was most likely near minor sites with high levels of contemporary land use that were proximal to more permanent sources of water (Fig. 2, Table 3). There was an interaction between land use and water such that the probability of finding plants was nearly 100% near permanent water under high contemporary land use. Dispersal predictors for *S. centrale* were consistently weak, as was predicted. Plants were weakly associated with ephemeral water in areas of high land use; however, wide confidence intervals make these results difficult to interpret. *Eragrostis* presence was also weakly explained by the dispersal model: the model predicted higher presence close to major sites near more ephemeral water sources that were under high contemporary use. Contrary to our expectations, we did see weak effects of dispersal proxies among *Scaevola*, but in the opposite direction as *S. diversiflorum*: scaevola was farther from sites, most often occurring farther from minor sites with less permanent water. However, the confidence intervals around these estimates are very broad and uncertainty is high. Overall, the fixed effects of the dispersal model explain a substantial amount of variation in the presence of *S. diversiflorum*, and the least among the other three species.

In step 2, we followed a model selection procedure (see Methods) to find the best predictive model for the presence and abundance of each species. Additional covariates were included in the model selection procedure to control for soil conditions, time since fire, season of last fire, water availability, strong interactions, and other differences between survey plots. The top model for each species was significantly better (P < 0.001) than a model including only random effects (Likelihood ratio test, SDIV$\chi^2$ = 131.84, SCENT$\chi^2$ = 69.94, ERAG$\chi^2$ = 307.04, SCAV$\chi^2$ = 53.66). In general, results from the best presence model corroborated the results from the global model with two exceptions (Tables 4 and 5). First, in the best *S. diversiflorum* presence models, the engineering covariates lost much of their predictive value when combined with dispersal covariates, and many dropped out of the best model. Land use remained strongly positive, and site distance strongly negative. Water permanence and site type remained important. Secondly, *Eragrostis* presence exhibited complex interactions between dispersal and engineering covariates that were only clear when we constructed the best model. For example, the negative effect of site distance was much stronger when we controlled for interactions between distance, land use, and water permanence.

The abundance models predicting stem counts when present performed poorly more often than those predicting presence, with wide confidence intervals, low pseudo $R^2$ (Nakagawa) values, and most of the variance explained by individual-level random effects (Table 6, Fig. 3). Even so, likelihood ratio tests indicated that the top model was significantly better (P < 0.01) than one containing only random effects (SDIV$\chi^2$ = 42.52, SCENT $\chi^2$ = 19.07, ERAG $\chi^2$ = 183.96, SCAV $\chi^2$ = 68.01). The most robust results predicted *S. diversiflorum* abundance to be highest close to sites near permanent water, and *Eragrostis* to be at higher abundance closer to sites with ephemeral water and more intensive land use.

## Discussion

Our results provide compelling evidence that highly mobile peoples living in one of the most arid environments in Australia acted (and continue to act) as agents of seed dispersal. This dispersal behavior has had a substantial impact on the distribution of culturally significant non-agricultural plants.

Martu people are likely to have been, and continue to be, significant dispersal agents of *Solanum diversiflorum*. The fruit occurs in dense patches and foraging returns are substantial. It continues to be an important component in the contemporary diet of Martu and their creation narratives, with the seed dispersed in such a way as to facilitate seedling growth in new and suitable areas. Ethnographic observations of *S. diversiflorum* harvesters, and surveys of food consumption sites, reveal substantial seed dispersal during harvest, transport, and processing that results in the plant successfully establishing new populations. As predicted in this study, plant abundance and presence shows evidence of both contemporary and past landscape use, being more likely to be present and at higher density in areas heavily used for contemporary hunting activities that are closer to minor occupation sites and near permanent water sources.

**Table 4 | Hypothesis test outcomes from the best model predicting presence and abundance**

| | Diversiflorum | | Centrale | | Eragrostis | | Scaevola | |
|---|---|---|---|---|---|---|---|---|
| | Presence | Abundance | Presence | Abundance | Presence | Abundance | Presence | Abundance |
| **Dispersal proxies** | | | | | | | | |
| Site distance | Strong negative** | Negative** | Weak negative under high land use** | No effect | Negative under high land use** | Negative near ephemeral water** | Weak positive | Weak positive |
| Land use | Strong positive** | No effect | Weakly Positive under high use** | No effect | Weak Negative | No effect | No effect | No effect |
| Water permanence | Strong positive** | Strong positive** | No effect | No effect | No effect* | Negative close to sites** | No effect | No effect |
| Site type Major | Negative** | No effect | No effect | No effect | No effect* | No effect** | Negative | Positive |
| **Engineering proxies** | | | | | | | | |
| Season Winter | No effect* | No effect | No effect | Negative | Negative | Positive** | No effect | No effect |
| Fire frequency | No effect | No effect | Strong positive* | Weak negative | No effect | No effect | No effect | No effect |
| TSF diversity | No effect* | No effect | No effect | No effect | No effect* | No effect | No effect | No effect |

No effect = coefficient not in best model: dropping coefficient reduced AIC value of the model during model selection and ranking. Positive effect = covariate increased presence or abundance. Negative effect = covariate reduced presence or abundance. *The predictor was significant in the global model but not in the best model. **Support for anthropogenic dispersal or anthropogenic engineering on presence or abundance.

**Table 5 | Best presence model results**

| | Diversiflorum | Centrale | Eragrostis | Scaevola |
|---|---|---|---|---|
| Site distance | 0.11 *** [0.05, 0.25] | 0.12 * [0.02, 0.61] | 1.13 [0.78, 1.64] | 2.32 ** [1.25, 4.30] |
| Land use | 6.37 ** [1.95, 20.81] | 0.59 [0.17, 2.00] | 0.49 * [0.28, 0.87] | 0.61 [0.25, 1.48] |
| Site dist × Land Use | 0.00 *** [0.00, 0.01] | 0.09 * [0.01, 0.67] | 0.03 *** [0.01, 0.09] | 11.82 ** [2.69, 52.07] |
| Water permanence | 13.73 *** [4.78, 39.42] | | 0.63 [0.28, 1.42] | 0.15 * [0.03, 0.74] |
| Fire frequency | 0.57 [0.27, 1.22] | 2.93 * [1.19, 7.18] | 0.72 [0.43, 1.22] | |
| Site type Major | 0.08 ** [0.01, 0.41] | | | 0.16 ** [0.04, 0.61] |
| Season Winter | | 0.28 [0.03, 2.37] | 0.27 *** [0.19, 0.40] | |
| Time since fire | | 0.82 [0.00, 39165.03] | 0.00 *** [0.00, 0.01] | |
| Season × TSF | | 0.00 [0.00, 2.62] | | |
| Land Use × Season | | | 3.45 *** [2.08, 5.71] | |
| Land Use × Water perm | 27302.32 *** [1282.01, 581445.45] | | 11.60 *** [3.04, 44.20] | |
| Site dist × Season | | 3.13 [0.63, 15.47] | | |
| Site dist × Water perm | 23.88 *** [4.04, 141.09] | | | |
| Season × Fire Frequency | | | 1.81 [0.90, 3.61] | |
| Carbon | | 0.11 ** [0.03, 0.46] | | 0.36 * [0.15, 0.84] |
| Total Veg Cover | | 3.63 * [1.35, 9.72] | 0.57 ** [0.40, 0.80] | |
| Elevation | | | 0.19 ** [0.07, 0.52] | 6.66 ** [1.80, 24.62] |
| NDMI | | | 1.73 * [1.11, 2.70] | |
| GreenVegCover | | | 0.64 ** [0.47, 0.86] | |
| PCT sand | 0.48 * [0.24, 0.98] | | 2.52 *** [1.62, 3.90] | 2.49 * [1.16, 5.38] |
| $R^2$ Fixed effects | 0.52 | 0.29 | 0.30 | 0.32 |
| $R^2$ Fixed + Random | 0.55 | 0.61 | 0.59 | 0.75 |

Model covariates (GLM, binomial, $n$ = 10 random effect levels, $n$ = 2924 observations) as the standardized odds ratio (2sd) for the probability of plant presence per plot (±95% CI in square brackets). Odds ratios higher than 1 indicate the predictor variable increases the odds of presence; those less than 1 indicate a negative relationship (the predictor variable decreases the likelihood of presence). Interaction (crossed) terms in the model are represented by "X". Caution must be used in interpreting the coefficients from interaction terms. $R^2$ terms are reported as Nakagawa's $R^2$ and are suggestive of the proportion of variation explained by the fixed terms, vs the fixed terms plus the random effect of Transect. Asterisks indicate significant $p$ values (* = <0.05, ** = <0.01, *** = <0.001).

People are also likely to be dispersal agents of *Eragrostis* grasses, although the relationship seems less straightforward than in the case of S. *diversiflorum*. These grasses are more likely to be present and at a higher density closer to sites with ephemeral water. This may reflect the different uses and seasonality of permanent *vs* more ephemeral water sources in the region. Ephemeral water sources were utilized in the wet season, when seed grasses ripened. Large ritual aggregations occurred primarily around ephemeral water sources such as claypans[39]. Seed cake consumption was (and continues to be) an important component of these rituals, especially for those involving male initiation. With seed grasses critical to successful conduct of ritual business, and occupation occurring mainly during the ripening period, there should be more processing of seed and more opportunities for heavy dispersal events, leading to more seed dispersal near ephemeral water sources. Permanent water sources were more likely to be used by smaller groups of people over longer time periods in the dry season, usually outside of the narrow seasonal window for harvesting *Eragrostis*.

## Table 6 | Best abundance model results

|                          | Diversiflorum              | Centrale                   | Eragrostis                  | Scaevola                  |
|--------------------------|----------------------------|----------------------------|-----------------------------|---------------------------|
| Site distance            | 0.21 *** [0.11, 0.41]      |                            | 1.63 [0.90, 2.96]           | 2.05 * [1.05, 4.00]       |
| Water permanence         | 4.59 *** [2.88, 7.32]      |                            | 0.84 [0.53, 1.32]           |                           |
| Site dist × Water perm   |                            |                            | 4.54 *** [1.99, 10.38]      |                           |
| Site type Major          |                            |                            |                             | 4.12 ** [1.75, 9.66]      |
| Season Winter            |                            | 0.24 * [0.07, 0.79]        | 2.09 *** [1.48, 2.95]       |                           |
| Fire frequency           |                            | 0.18 ** [0.06, 0.52]       | 1.40 [1.00, 1.97]           |                           |
| Season × TSF             |                            | 0.00 * [0.00, 0.69]        |                             |                           |
| Season × Fire frequency  |                            | 6.30 * [1.21, 32.88]       |                             |                           |
| TSF                      |                            | 32.15 [0.15, 7072.10]      | 0.01 *** [0.00, 0.05]       | 0.07 ** [0.01, 0.49]      |
| Soil Carbon              | 0.50 ** [0.33, 0.78]       |                            | 0.31 *** [0.19, 0.49]       | 0.21 * [0.06, 0.70]       |
| PCT Sand                 | 0.41 ** [0.21, 0.79]       |                            | 1.31 [0.96, 1.80]           |                           |
| PCT Veg cover            |                            |                            | 1.92 *** [1.34, 2.75]       |                           |
| NDMI                     |                            |                            | 1.60 * [1.11, 2.31]         |                           |
| TSF × Site distance      |                            |                            | 9.04 * [1.44, 56.76]        |                           |
| N (observations)         | 109                        | 70                         | 617                         | 177                       |
| R² Fixed effects         | 37.81%                     | 25.60%                     | 27.50%                      | 33.90%                    |
| R² Fixed + Random        | 82.75%                     | 92.68%                     | 98.70%                      | 99.90%                    |

Model coefficients (GLM, Poisson, observation level random effects) as the standardized odds ratio (2 sd) for the best presence models (±95% CI). Odds ratios give the relative change in the likelihood of presence for a two-standard deviation increase in the predictor variable. Odds ratios higher than 1 indicate a positive relationship (the predictor variable increases the likelihood of presence); those less than 1 indicate a negative relationship (the predictor variable decreases the likelihood of presence). Interaction (crossed) terms in the model are represented by "X". As is common in Poisson models, we account for overdispersion through the inclusion of individual-level random effects. Asterisks indicate significant $p$ values (* = <0.05, ** = <0.01, *** = <0.001). Predicted counts for some variables and interactions are presented in Fig. 3.

People are less likely to be dispersal agents of *S. centrale;* however, they may influence populations through changes to the fire regime. S. *centrale* presence is more likely closer to sites, and it is also strongly affected by fire frequency, likely through stimulating clonal reproduction[40]. Together, this suggests weaker effects of dispersal and stronger effects of landscape engineering. This is as expected given the edible nature of *S. centrale* seeds, the strong dependence of this species on vegetative reproduction, and the higher potential for dispersal by other non-human consumers.

As predicted, our control species, *Scaevola*, exhibited no effects of dispersal and no positive effects of landscape engineering. Patches were more likely to be present (and denser) far from occupation sites. This may reflect some degree of interaction between people and at least two of the major dispersers of *Scaevola*, the Australian bustard (*Ardeotis australis)* and the burrowing bettong (*Bettongia leseur*). Martu have noted that bustards and other prey animals will avoid sites frequented by people due to fear of predation, with the consequence that plant dispersal around occupation sites may be suppressed.

That plants are more likely near old habitation or archeological sites suggests either that contemporary visitation has resulted in seed being dispersed here, or that there are substantial historical legacies of prior fruit processing, or (we suspect) both. While some archeological sites have not been utilized for upwards of 70 years, most of those within the survey region are still frequently visited for both day trips and for longer term camps. Even if the association between sites and plants are the product of pre-contact plant dispersals, this would not be too surprising: long-term effects have been observed in other social-ecological systems, but generally not among populations with such high residential mobility[7,41]. Yarnell[9] was one of the first to recognize that village sites among agriculturally-based populations in the American Southwest were more likely to host wild plants of significance to local Navajo and Pueblo peoples. In his work, he specifically noted that five species of Solanaceae were not grown in gardens yet were more common close to archeological sites. More recent work has established a strong relationship between site complexity (e.g., the number of elements) and the presence of culturally important 'wild' plants, including *Solanum*, seed resources, and fruit with inedible

seeds[7], as well as human-induced founder effects visible in *Solanum* genetic diversity[41]. In the pacific Northwest, important "wild" food plants are concentrated near archeological sites in patterns that are suggestive of active tending[11,42,43]. Modifications of wild plant distributions have also been documented as a product of pastoralist economies: some are more abundant near former corral sites in both the arid SW and in the south-central Andes[44,45]. In this case, however, the mechanism of dispersal lies in mutualisms between domesticated animals and the plants they consume. One of the few studies to document effects of highly mobile hunter gatherers on plants comes from central Africa, where wild yams (*Dioscorea* spp.) are found at higher densities near Baka camping places, as a product of the inadvertent discard of yam tops, which propagate in soil disturbed by human bioturbation[46,47]. Our study suggests that even under mobility patterns where the nature of occupation at sites is intermittent, and populations are at very low density, anthropogenic seed dispersal can have a lasting and observable effect on plant communities for decades, if not much longer.

Recognition of the often unintentional dispersal or disturbance mechanisms that shape ecological communities can improve studies of the formation of cultural landscapes, which often assume simply that traditional ecological knowledge directly produces ecological outcomes (e.g., refs. 15,48,49). However, knowledge cannot have a material impact without practice, and while many practices are cognized, others may be more implicit, or even have unintended ecological outcomes. The relevance of indigenous knowledge for sustainability and conservation would only strengthen if we better understand the processes through which knowledge informs specific practices, and how those practices then shape landscapes, regardless of whether such shaping is intentional management or due to other non-cognized process.

If people can create cultural landscapes simply through the way they interact with plant propagules[50], debates such as the one over whether the First Australians were farmers or foragers[51,52] become moot. Seed dispersal figures prominently in the debate, with Pascoe[51] arguing that historical evidence of deliberate seed scattering constitutes farming. In a detailed critique, Sutton and Walshe[52] recast seed

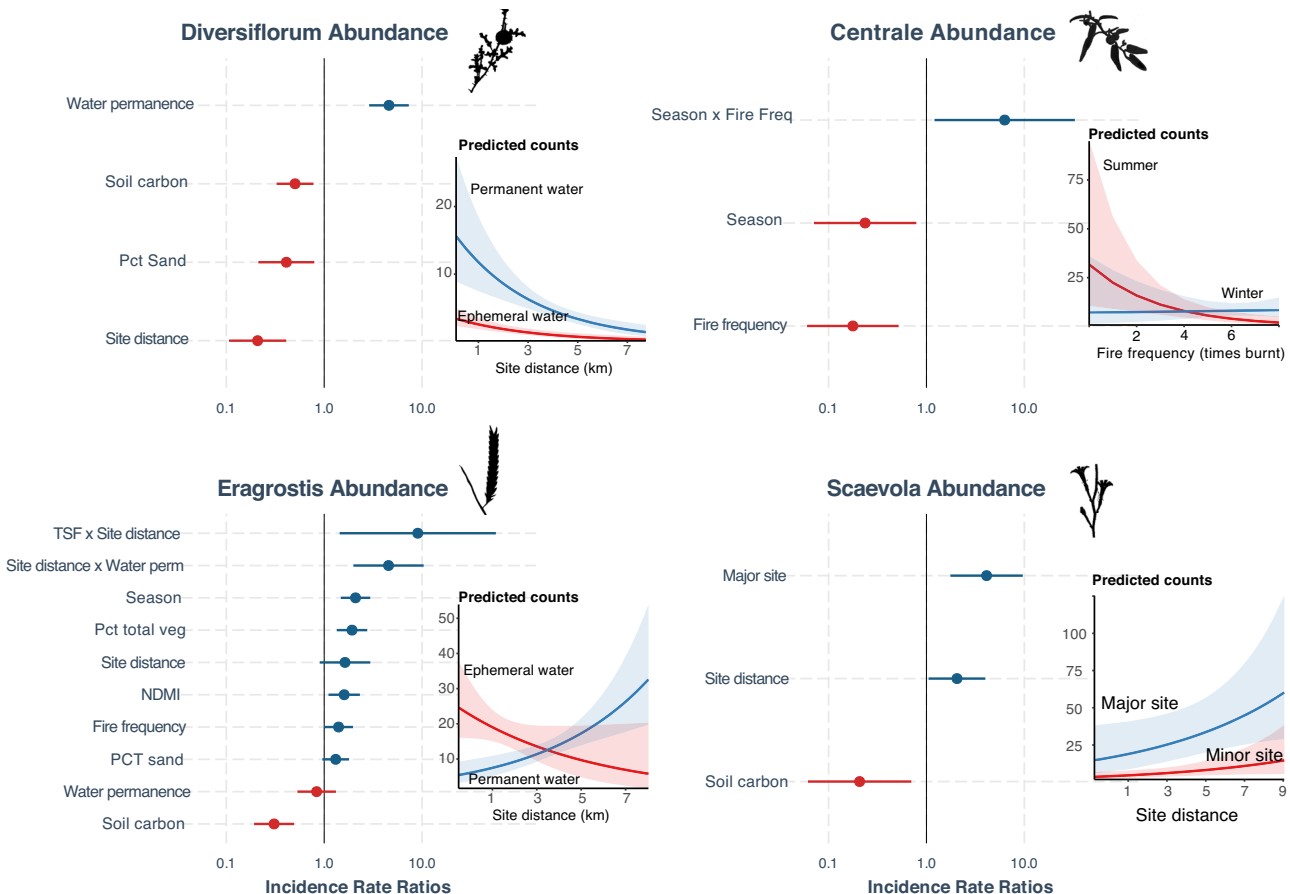

**Fig. 3 | Incident rate ratios for predicted plant abundance.** Dot and whisker plots on the left show the strength of each predictor variable as an incident rate ratio with 95% CI for the best abundance model for each species (see values in Table 6). Red values are negative (below 1), blue are positive (above 1). Coefficients have been standardized to 2 sd, so that an IRR of 0.5 means that a 2 sd increase in the value of the predictor decreases stem counts by two times; an IRR of 2 increases stem counts by two times. Confidence intervals for IRRs that cross the line at 1 are non-significant. IRRs for interaction terms (e.g., Season × Fire Freq) are to be interpreted with caution. Since this analysis predicts stem counts only when plants are present in the plot, sample size (plots present) for each species varies: SDIV(n) = 109, SCENT(n) = 709, ERAG(n) = 617, SCAV(n) = 177. The inset line graph (right) shows the predicted abundance (with 95% CI) and interactions for the strongest predictor.

dispersal occurring as part of ritual practices designed to ensure plant regeneration. They believe such practices were misconstrued by early observers (mainly explorers, pastoralists and missionaries) as intentional seed dispersal[53]. We argue that while there is occasional scattering of seed that is intended to produce plants in particular places, most seed dispersal is shaped by other intentions, particularly processing and transport decisions. Although these decisions are epiphenomenal to later plant growth, they still matter in affecting food availability in the future—demonstrating the inadequacy of the categories of foraging and farming to account for these unintentional processes of seed dispersal. Seed dispersal has significant ecological outcomes regardless of whether it is "intended" or not (e.g., ref. 50).

Knowing which plants are strongly affected by indigenous seed dispersal may also better inform strategies of conservation and restoration, especially for rare, wild plants. In our study, each species was affected differently by interactions with people: disturbance through fire was more important for *S. centrale*, seed dispersal was more critical for *Eragrostis* and *S. diversiflorum*, and (possibly) potential suppression of nonhuman dispersers for *Scaevola*. Given that many native *Solanum* species may be threatened by habitat loss and changes to fire regimes[14,54–56], a recognition of the past Aboriginal role in burning, seed dispersal and the promotion of genetic diversity is integral toward the design of better restoration and conservation plans for these rangelands. A lack of recognition of the role people can play in mutualistic interactions with other species contributes to the

shifting baselines problem[57] in conservation and restoration ecology, when those interactions are no longer present. If we assume the baseline is a landscape devoid of people, when in fact the landscape has been occupied and fired for millennia, we risk setting conservation goals that maintain species in an impoverished state relative to their historic potential. If we assume that the processes that maintain population stability only involve non-human agents, we risk implementing restoration or conservation protocols that are sub-optimal. If policy frameworks only consider a single mechanism influencing interactions with other species (e.g., fire), we will miss a wide variety of other types of ecological interactions (e.g., seed dispersal, predator suppression), some of which favor some species and disfavor others. A more nuanced understanding of human agency and practice in creating legacy ecosystems is crucial to the informed and sustainable management of conservation estates into the 21st century.

## Methods
### Plant dispersal mechanisms
Bush tomatoes are a highly desirable cultural resource for Aboriginal people living in remote desert communities. Not only is the fruit a high quality source of nutrients and calories that can be obtained with relatively little effort, it also plays an important role in people's ritual lives as expressed through the Dreaming[36]. For Martu, there are two species of primary importance: (a) *Solanum diversiflorum* (known locally as *wamula or kumpulpaja*), a sweet, large-fruited species (25 g)

with a hollow interior holding a mass of inedible, bitter, black seeds that must be removed before consumption (Fig. 1); and (b) *Solanum centrale* (*jinjiwirri*) a small-fruited species (5 g) with edible seeds embedded in a fleshy, sweet berry. Both species are more abundant following recent fire[19], and in our study, more than 90% of observations of plant presence occur within 5 years following a fire.

Because *S. diversiflorum* has fairly large and inedible seeds encased within heavy fruit, it is most likely to be dispersed via epizoochory (active or passive transport of seed), in which people carry fruit to a location for processing. Seed dispersal is thus likely to be associated with places on the landscape where people have camped, in both past and present. People are likely to be the most important seed dispersers, as it is unclear what animals beyond humans regularly consume *S. diversiflorum* fruit, or act as its dispersal agents[58–60], and it appears there is little competition with people for the fruit. Most fruits that go unpicked by people simply dry up on the stem and the seeds drop to the base of the plant within the desiccated fruit case. On rare occasions, Martu have pointed out *S. diversiflorum* fruit that hill kangaroo have nibbled, but the kangaroo do so in a way that leaves the seeds attached to the calyx hanging from the stem—they do not transport seed beyond the parent plant.

In comparison, *Solanum centrale* is less likely to be dispersed primarily by people. The seeds are small, edible and consumed rather than discarded; the fruit is small, soft and is always transported whole to residential sites or eaten on the spot; historically, it was often processed into large cakes and dried for storage and transport[61]. Some epizoochory through seed loss might be expected based on the occasional escape of whole or dried fruit from dishes and seedcakes as it is transported cross country, but we believe this occurred on a scale that was much more restricted than *S. diversiflorum*. Depending on the efficiency of digestion, human endozoochory (consumption and excretion of undigested seed) might be possible for this species, although Martu traditional knowledge does not recognize this as a potential dispersal mechanism. We do note that Solanaceae taxa with similar morphology (such as *Physalis*, which also possesses small fleshy fruits with embedded seeds) have been found at high densities in human coprolites[62]. People are not the only potential disperser of this plant, however, as it is often consumed by many other organisms, especially hare wallabies[63] (*Lagorchestes hirsutus*, now locally extinct) emus (*Dromaius novaehollandiae*) and bustards (*Ardeotis australis*)[64,65]. Aboriginal Dreaming narratives relate how birds often fight over the fruit. It rarely remains on the plant for long after ripening. Unlike *S. diversiflorum*, it is clonal; thus vegetative propagation may be a more significant influence on its distribution than seed dispersal[40,58], and past anthropogenic contributions to its dispersal may be difficult to detect. In addition, because people now typically bring most of the fruit back to the community for consumption, there are likely to be fewer opportunities for endozoochory at a landscape scale and thus we would not expect to detect strong effects of contemporary landscape use.

Our research design makes use of a controlled comparison not only between two Solanaceae that are more or less likely to be dispersed by people but also between two additional taxa: *Eragrostis setifolia/eriopoda* (both species of Eragrostis are utilized interchangably and not distinguished by Martu foragers), and *Scaevola parvifolia*. Both plants are common and co-occur with bush tomatoes in early-mid successional sandplain habitats (Fig. 1C), and both are edible but much lower ranked (yielding lower rates of energetic return due to high processing costs) compared with both bush tomatoes. Of the two, *Eragrostis* is more likely to be human-dispersed. It is a tussock forming grass locally known as woolybutt (*kunuruntu*), which produces a tiny edible seed nearly identical to the Ethiopian domesticate *teff*; it was historically significant in the diet but very low ranked (low net caloric gain relative to its processing cost[66]) and is not currently collected except on rare occasions. This species could respond to

contemporary human fire use and past foraging activities, as seed may have been dispersed near residential sites due to losses during transport and threshing/grinding activities, but such signals will be difficult to detect as the seed could also be dispersed by a wide range of other seed consuming organisms (ants), wind and water.

*Scaevola parvifolia* is least likely to be dispersed by people and as such, serves as our control species. It is a small herbaceous plant with an edible but very low-ranked tiny fruit about 0.5 cm in length and 1 g in weight, with a fleshy, gooseberry-like portion that comprises only a thin skin over a large, inedible internal seed. Often dismissed as *kipara mayi* by Martu people (bustard food), it is not eaten in any great numbers by adults, although children occasionally consume small amounts of fruit on the spot[28]. Formerly, it may have been consumed by the burrowing bettong, extinct in the region since the 1960s[67]. Of all four species, it is the least likely to be influenced by anthropogenic dispersal.

## Hypotheses and predictions

If *S. diversiflorum* is dispersed by people, the intensity of landscape use in either the past, the present, or both should predict presence and/or abundance (see Table 1). During the pre-contact era for Martu peoples, residential camps were situated near water sources of variable permanency depending on season[68,69]. If past landscape use has had a significant effect on plant presence, plants should thus be more likely to be found close to occupation sites and where the nearest water source is more reliable. However, when patches of fruit were more distant than a kilometer or two from camp, people would often make a temporary logistical or foraging camp near the patch to remove seed in bulk[68,70] before carrying the processed fruit back to the main residential site. Thus, for this species, the type of occupation site may also matter: this region is well surveyed, and hosts a wide range of site types covering the last several millennia[68,71]. Whether or not a site is used for short term occupation vs long term occupation could influence spatial variation in *S. diversiflorum* seed due to the way that people remove seed to increase the utility (proportion of load comprised of useful parts) of a transported load[48]. This is because processing of resources with inedible parts is subject to threshold effects, such that the distance one must move the item and the time to remove its inedible parts influences whether one will field-process the item or carry it home to be processed at camp. Because the processing time of *diversiflorum* is fairly quick (30 s per fruit) and the load utility (percent edible) increases dramatically with the removal of the inedible central portion (allowing the hollow fruit to be nested together), the transport threshold should be fairly short, meaning that people will be more likely to make a temporary camp near a fruit patch to clean and discard seed if the patch is farther than a kilometer or two from a water source. Patches should thus be more likely to be found near ephemeral or task-specific sites used mainly over the short term by smaller groups of people (minor site), compared to those sites with evidence of longer-term use by larger groups of people (major site).

Present landscape use should also predict presence/abundance. Plants should be more likely to be found or be at higher density in landscapes that are more intensively used for present day foraging activities. Our previous, systematic work on anthropogenic fire regimes has shown that the number of winter season fire footprints in a region is closely correlated with foraging effort, particularly for hunting monitor lizards (*Varanus* spp)[49]. If people are frequently hunting in a region, companion foods such as fruit may be more likely to be collected. Here hunting intensity as measured by fire density, is treated as a proxy for the general intensity of landscape use, which could be more accurate than travel distance from a central community, which we have previously used as a measure of the intensity of landscape use. Ethnographic observations of plant collection and processing should also provide evidence for the potential for seed dispersal, both past and present: places where *S. diversiflorum* was known to be processed in the past should have a higher probability of plant presence than nearby locations without such histories, and observations of foraging

activities should reveal systematic discard of seed around camping locations.

Our predictions concerning past and present landscape use for *S. centrale* and *Eragrostis* differ from that of S. *diversiflorum* since the dispersal mechanisms are different. For both of these plants, there should be no effect of site type, as there is no field processing of either species. Second, the effects of past land use as inferred by site distance and water permanence rank (as proxies) will be weaker and therefore more difficult to detect. Ethnographic observations should show the potential for seed dispersal for both species, however our measures of contemporary landscape use may have little, if any, effect on either. Our control species, *Scaevola*, should show no effect of past or present landscape use, and ethnographic observations should reveal little if any potential for direct seed dispersal.

While seed dispersal via epi- and endozoochory are potential mechanisms for influencing plant distribution, changing disturbance regimes, most obviously through fire, may also play a role for some species. By burning small grassland patches more frequently, anthropogenic fire regimes constructed and maintained by Martu people in the Western Desert shape habitat structure by increasing local pyrodiversity (the diversity of patches at different time since fire). This reduces the scale of habitat patchiness compared to landscapes where lightning dominates, and promotes the abundance of many native animal species[72]. This reshaping of landscape vegetative diversity is a type of ecosystem engineering for many plant species, which may reduce the cost to people of acquiring early-mid successional plant resources[73] and may also increase the availability of some culturally significant plant species[17,18]. Martu use of fire at a landscape scale creates an intermediate disturbance that could favor plant presence for all four species independently of seed dispersal mechanisms. Potential measures of anthropogenic landscape fire that distinguish human fire regimes at the landscape scale are time-since-fire diversity[74,75] and at the patch scale, the seasonality of firing events (most anthropogenic fires occur in the cool dry season, while most lightning fires are in the hot summer months). Fire frequency increases under anthropogenic influence in some ecosystems (e.g., SE Australia[76], but it is not clear whether it does in this case: prior work suggests that people increase the diversity of frequencies, e.g., increase both low and high fire frequencies at a landscape scale[75,77]. Here, we have no specific expectations for each species, but test against the null hypothesis of no effect of fire. While all our plants are clearly fire-dependent, lightning fires are a frequent source of landscape fire in this region, thus dependence on fire itself is not a clear indication of dependence on human fire regimes.

## Ethics and inclusion

Research approvals from the Martu Prescribed Body Corporate (now Jamukurnu Yapalikurnu Aboriginal Corporation, the trustee for Martu native title rights) were first obtained in June 2000, renewed in July 2004, and again in 2023. All ethnographic work was conducted under human subjects approvals from Stanford University (#1915, 2005-2015) and Penn State University (#00011455, 2015–present). Informed consent was obtained from all participants.

While intermittent interaction between some Martu and Europeans began early in the 20th century, others avoided contact with settler-colonial invasion until the 1960s. Many remote-living Martu were first contacted in the 1960s during government welfare patrols tasked with evaluating the drop zone for inter-ballistic missiles that were eventually launched in the International Weapons Research Establishment, Australia's principal contribution to Cold War efforts. The last groups left the desert in 1967, but reoccupation of Martu lands began in the 1980s with the establishment of three remote communities at Punmu, Parnngurr, and Kunawarritji. One of the authors who is Martu (DT) was born in Martu country before the hiatus and was involved in all aspects of the homelands movement; while another (PV)

began archeological programs in collaboration with remote-living Martu families in the mid-1980s. Many of the archeological sites recorded by PV with Martu custodians in the 1980s have been re-occupied with the advent of the outstation movement. Both pre-contact archeological assemblages and contemporary plant use around the sites, were recorded with ethnobotanists[28,68,69,78]. RBB and DWB began work here in 2000 as Martu were preparing their Native Title claim, which was determined in 2002[31].

On return to their homelands, Martu reinvigorated their customary subsistence and landscape burning practices[32,33], and many remote-living families continue to maintain fundamental aspects of their foraging livelihoods, typically spending about 25% of their days in hunting or gathering activities[28,36,79–84]. People often go out foraging on day trips to various locales, which includes establishing a temporary 'dinner camp' for the purpose of bush food preparation, cooking and consumption. We have previously reported quantitative details of foraging time allocation, labor decisions, bush food yields, return rates from regular hunting and gathering practices, the role of cultural burning in facilitating hunting strategies, the ecological role of these practices, and significant changes in subsistence over the last half-century[32,33,36,3781–86].

While the three remote communities in the Martu Native Title lands have semi-permanent infrastructure—including housing, a primary school, a small shop, and a health clinic—it would be misleading to describe Martu as "settled" or even "semi-settled". Most exercise extremely high mobility, shifting residence many times over the course of a year across the remote communities and throughout settlements and towns beyond the Native Title boundaries. The composition of Martu residential communities and foraging groups is thus constantly fluctuating[87]. Their mobility and foraging activities are vital in maintaining the landesque capital of their estates, fundamentally entangled in a web of both social and ecological interactions between people, fire, plants, and animals that make the landscape of country "ngurra-ra", or homeland[33]. These interactions are material manifestations of the creative epoch (The Dreaming), in which people re-enact the process of creation through the maintenance and support of ecological networks[32,52,86,88].

The ethnographic and ecological work reported in this study began in June of 2000 by DWB and RBB as part of a long-term project centered on understanding the socio-ecological dynamics of human–environment interactions. As part of this project, we quantitatively recorded observations of time allocation, production, distribution, and cultural burning associated with a broad set of foraging activities and land use by Martu men and women. The foraging data used in this study were obtained by accompanying a foraging party leaving the community over a cumulative 800-day observation period between June 2000 and August 2017. Martu typically use vehicles to access the day's foraging region. On arrival they establish a temporary foraging camp (referred to as *mirrka ngurra*) from which people usually depart on foot, solo or in small cooperative groups. Depending on season and habitat, women most often engage in burning patches of older growth spinifex grass to facilitate their monitor lizard hunting, while men focus much of their efforts in searching for bustards or hill kangaroo. After the day's foraging, people gather back at the hearth of the foraging camp to process, cook, share, and consume the day's catch before returning to the community in the evening[32–34,36,,81,82,84,85,89,90]. During foraging trips, we recorded all foraging time allocation, yields, and distribution of bush foods among all participants, along with focal follows of individual foragers recording all search, pursuit, capture, transport, and processing. These data comprise a total of 385 trip-days averaging 8 people per trip and 160 unique individuals (80F, 80M). We recorded *Solanum* harvesting time allocation, yields, and sharing on a total of 28 of these trips, although we participated in many more *Solanum* harvests than were recorded quantitatively.

## Plant presence surveys

Occurrence records[91–94] for all four species place the study area well within each species' known range (Fig. 1C), with the caveat that our study area is very remote, roadless, and poorly sampled. Our own surveys show all four species to be common and widely distributed in spinifex sandplain habitat across the entire study area. Plant presence was assessed using pedestrian surveys of plant distributions conducted in June and July of 2003. These surveys employed ten 10 km × 10-m belt transects to assess presence, absence and abundance via stem counts (see Table 2). Transects radiated outward in two directions from locations centered at ten past and present camping places so that we could more accurately capture the effect of distance from site on plant presence and abundance. Transect location was also stratified according to landscape use intensity, with two transects near ethnohistoric sites in little used landscapes, two near sites in heavily used landscapes, and the remainder near sites in landscapes with moderate use. Stem counts (counting main stem only) were aggregated to 300 m² in the analysis to more closely match the 30 m pixel resolution of the remotely sensed imagery, producing a total of 3130 plots of which 2924 were located in early-mid successional habitat.

We also assessed the potential for contemporary Martu fruit processing to result in newly established patches of *S. diversiflorum* fruit by surveying previously observed fruit consumption sites for the presence and abundance of *S. diversiflorum* (Fig. 1G). In 2018, we selected 12 foraging locations with central hearths used between 2002 and 2017, at which bush foods were consumed, and for which we had complete quantitative foraging records. Half of these sites (controls, $n = 6$) had no *S. diversiflorum* consumption recorded during use. At the 6 remaining sites we had detailed records of *S. diversiflorum* harvesting, consumption, and seed dispersal, and the hearth was located in an area where no plants were growing at the time of use. Each foraging camp dispersal survey consisted of one 50 × 50 m plot centered at the cooking hearth around which people were sitting as they ate, and one randomly placed comparison plot located at least 500 m from the hearth. To avoid conflating present and past land use effects, none of these survey sites was located near archeological or ethnohistoric sites as described below.

## Measures of past landscape use: distance to ethnohistoric site, water permanence, and site type

To operationalize questions about how past legacies of land use may shape landscape-level distribution of key plants, we ask whether the locations of contemporary plant patches are affected by distance to both ethnohistoric and archeological sites[7]. Ethnohistoric sites are sites that are known in living memory as old residential sites; archeological sites are those that have a known archeological component whether or not people remember using them as residential sites in the past. We do not distinguish in our analysis between the two types of sites. Before their homeland exodus, Martu utilized hundreds of residential campsites and foraging camps within the study area. While visits to more remote sites declined during the mid-20th century hiatus, Martu resumed frequenting many of these sites upon returning to their homelands. Most of the sites in the study area are within daily travel range of the remote communities and are still utilized for both short-term foraging camps and longer multiday camping trips. While the earliest use of most of these sites (ethnohistoric and archeological) is not known, the depth of Martu ancestral ties in this region is profound, with occupation of these landscapes extending well into the Late Pleistocene as far back as 48k years ago[30,95]. The sample of sites in this analysis were all in use prior to Martu exodus from their homelands in the 1960s, meaning that all have had at least some use within the last 70 years, and many still continue to be used in the course of regular foraging activities.

To evaluate the legacy effect of anthropogenic land use on contemporary plant distribution, archeological site locations from publicly available datasets ($n = 192$, see refs. 70,72,96) were supplemented with informant recall of residential and foraging camp locations ($n = 179$ sites), and water sources known on topographic maps likely to be used as camping places ($n = 21$ sites). To capture any missing and potentially significant residential sites, we also surveyed high-resolution satellite imagery within a 20 km radius of the survey location for any significant water sources not already recorded, mainly soaks, springs, pools and larger rockholes. Using expert knowledge, including our own on the ground observations, each water source was associated with a relative permanence value, 1: ephemeral, to 4: nearly year-round availability. Camping sites and potable water sources are highly correlated: out of 192 sites with known archeology, the median distance to the nearest water source was 296 m, and 90% of all sites in our database are within 3 km of water. We are confident in assuming that water sources for which little information is recorded are likely to be potential camping places. To cross-check the campsite surveys, we used an aerial photo mosaic from 1953 which provided coverage for 366 campsite locations. 86% of sites showed signs of use in the form of nearby anthropogenic fire mosaics. We employ those historic photo mosaics and interviews with elders to assist in classifying each site according to its type of use. Major sites ($n = 74$) were defined as such based on ethnographic knowledge of the location as a major meeting place, ceremonial site, or aggregation site and/or visible signs of extensive fire mosaics in 1953. Where detailed archeological mapping, sampling and dating (both absolute and relative) had occurred[68] this was cross-checked with information from Martu elders. The remaining sites were assigned as minor sites ($n = 376$). Informant knowledge of such sites is likely to be biased to those locations frequented by the few people to remain in the desert post 1950. The only potential sites missing from this analysis would be those which are associated with water sources invisible on satellite imagery that were abandoned prior to the 1950s and for which there is no ethnographic or archeological knowledge. However, given that the plant sampling transects were solely within regions well known to informants and to many of the authors, it is unlikely that this bias affects our results. To obtain the distance to nearest ethnohistoric site from each survey location, we performed a nearest neighbor analysis in ArcGIS Pro v. 2.5[97]) on the center points layer of each 30 m transect interval and the center points of the ethnohistoric site layer.

## Measure of present landscape use: winter fire density

We measured hunting intensity by summing the density of remotely sensed winter season fire footprints as visible in Landsat imagery over a 30-year period prior to the survey date (1973–2003). We reconstructed the fire history of the study region using a time series of 75 30-m resolution Landsat 2–7 image mosaics taken at roughly 6 month intervals (October and April) between 1980 and 2003. Two additional time steps were available from 1973 and 1979, which were used to attach an approximate age to the fires visible in the 1980 base image. Between 1987 and 2003, surface reflectance normalized burn ratio imagery was available from USGS-ESPA, from which we constructed difference images to highlight only those fires burning in each time step. Between 1973 and 1987, fires were detected using the difference of band 4 between time steps. Fires were hand-drawn using the pixel-based region-growing algorithm in QGIS 3.14[98] and each time step was converted to a raster image. For our landscape use proxy, only winter fires were counted as there are no non-anthropogenic sources of fire during that season. Winter fire footprints are small and easily represented by a single center-point; we converted the point layer to a raster fire density layer using a kernel density estimation with a 3 km radius. We then extracted fire density for each presence point by using a point sampling function to query the raster layer.

## Fire covariates: time-since-fire, time-since-fire diversity, fire frequency, proportion of ground cover

Fire regimes were assessed using the Shannon diversity index of TSF age classes, which are calculating using remotely sensed 20-year fire

history maps[74]. To construct the fire history maps in QGIS 3.14, we stacked each raster fire footprint between 1983 and 2003 with the most recent fires on the top layer. Each layer was given a value corresponding to the elapsed months since time zero in 6-month intervals (with Oct–April 1983 being time zero), such that the resulting image maps out the time since last fire of each landscape patch. Martu landscape use has a direct impact on Shannon diversity measures, increasing diversity close to communities where people forage and burn more actively[74]. Shannon diversity was measured at the 3 km scale in a circular region centered on each 30 m plot centroid as this scale best differentiates anthropogenic fire regimes[49].

All four plant species also exhibit declining density with time-since-fire (see Table 2). TSF for each transect plot was assessed in two ways: on the ground, by experienced researchers estimating months since last fire, and secondly, by utilizing this ground classification in conjunction with fire history maps to convert the categorical classification and estimations into months since last fire. As the resolution of the satellite imagery used to derive the fire maps was a maximum of 30 m, and the classification maps were at best only 90% accurate, there were some discrepancies between ground and satellite-based estimation of TSF, especially at the fire boundary; these were resolved by using the ground estimation to adjust the boundary of each TSF age class.

Plant presence may also be affected by the frequency of fire, which is influenced by the interval between fires. Martu burning increases the diversity of fire frequencies, creating patches close to the community where fire is both more frequent and less frequent[75]. Fire frequency at the time of survey was estimated using the fire history maps, as the number of times burnt in the 30 years prior to the survey.

Time since fire is only a rough proxy of successional vegetation structure, as rainfall controls the rate at which plants grow after fire[99,100]. To account for this, we also added a covariate describing the relative proportion of green, brown and total vegetation sampled at each plot centroid. Green vegetation increases with rainfall and is dominant soon after fire, while brown vegetation consists primarily of senescent *Triodia* grass hummocks and dominates 5–10 years after fire. These raster layers were created in QGIS 3.14 from Digital Earth Australia 30 m Landsat-derived fractional cover datasets available through the Terrestrial Ecosystem Research Network[101].

### Soil covariates: soil moisture, rainfall, soil components, slope, and aspect

Soil water limitations combined with microgeological variation in nutrient availability are two fundamental drivers in the distribution of endemic plant species across Australia[102]. To help control for this variation in water and nutrient availability, we included several topographic elements (e.g., slope and aspect) and soil attributes (e.g., water-holding capacity, clay content and sand content) within our analysis. For the terrain features, we derived slope (degrees) and aspect (the compass direction of the slope) values at each sample location in ArcGIS Pro 2.5 from a 1 s SRTM digital elevation model (DEM, resampled to 30 m) provided through the USGS. Both slope and aspect have been shown to influence the distribution of plants in semi-arid regions through the creation of microclimates (e.g., soil temperature, evapotranspiration, wind speed, etc.), the alteration of soil properties (e.g., organic matter context, soil depth, texture, etc.), and the control of hydrological processes (e.g., runoff dynamics, soil water retention, etc.)[103,104]. Soil properties were available as a set of 30 m raster layers available through the Soil and Landscape Grid of Australia[105]. Those that we included in the initial parameter selection were organic carbon, available water capacity at 0–5 cm, percentage clay, silt, phosphorus, and sand.

To capture variability due to persistent soil moisture, we calculated NDMI (the normalized difference water index, a measure of vegetation moisture) as a long-term average measure (since 1987),

using 30 m Landsat imagery available as a derived product from the EROS data archive. We also used cumulative rainfall for the 12 months prior to the survey date, drawing on 5 km scale gridded (raster) monthly rainfall datasets available from the Australian Bureau of Meteorology. The grids were resampled in QGIS 3.14 to 30 m, summed over the relevant 12-month period for each survey date and the total rainfall amount at each survey center point was extracted using the raster point sampling tool.

### Analytical methods

We explore the anthropogenic and environmental factors influencing species distributions with an information theoretic model selection approach[106] in R (v 4.2.2)[107], predicting plant presence/absence using generalized linear mixed models (glmer) for the binomial distribution in the lme4 (v 1.1) package[108]. Because we had many, often collinear, non-anthropogenic climatic and geophysical covariates that could obscure the relationship with our anthropogenic predictors, our variable selection approach followed best practices outlined in ref. 109. We first screened the potential set of non-anthropogenic covariates for associations with the presence of each species, and then checked for any significant collinearities. This initial process identified a total of nine of the most informative, non-collinear soil/climate and fire-related covariates: soil carbon, percentage green ground cover, percent sand, long term average NDMI, elevation, percent total green+brown vegetation, time-since-fire, fire frequency since 1973, and season of most recent fire (winter vs summer). These were retained, along with our anthropogenic predictors, in the model selection approach. All binomial presence models included a random effect (transect, $n = 10$) to control for spatial covariance in the transect plots; poisson models for abundance included individual-level random effects as recommended to reduce overdispersion[110]. After running each model and checking model diagnostics for spatial autocorrelation and fit, we used the MuMIn package (v 1.47.1) in R to calculate AIC, BIC, model weights and Nakagawa's pseudo $R^2$ values[111].

The model selection procedure to identify the best models for both presence and abundance involved four steps. First, we ran models predicting species presence with only the random effect, then with each single covariate + random effect. Secondly, we combined all variables of each type (anthropogenic, fire, and soil) into three separate global models (considering only first order interactions), removing the poorest performing covariate until we reached a minimum AIC value. Third, we explored combinations of anthropogenic and fire variables, fire and soil variables, and anthropogenic and soil variables. Fourth, we combined all covariates and predictors into a single model, again removing the poorest performing until we reached a minimum AIC value. Details of all model parameters, AIC values and Akaike weights can be found in Supplementary Data 1.

### Reporting summary

Further information on research design is available in the Nature Portfolio Reporting Summary linked to this article.

## Data availability

Source data obtained on Martu Native Title (plant abundance surveys, archeological site and water source locations, foraging focal follow observations, fire history mapping) are available under restricted access due to the need for each user to sign intellectual property agreements with the Jamukurnu Yapalikurnu Aboriginal Corporation. Data will be archived in Penn State Scholarsphere (https://scholarsphere.psu.edu) and access may be obtained by request of the corresponding author after consultation with JYAC. The remainder of the data used in this analysis is derived from publicly available datasets including the Soil and Landscape Grid of Australia and Landsat-derived vegetation fractional cover datasets from the TERN repository (https://portal.tern.org.au); the SRTM digital elevation model from the US. Geological Survey (https://

earthexplorer.usgs.gov), and archived rainfall data available through the Australian Bureau of Meteorology (https://bom.gov.au). Occurrence records (not including Martu-owned data) for all four species are available through Atlas of Living Australia (https://www.ala.org.au).

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

## Acknowledgements

This research was supported by NSF grants BCS-1917937 (R.B.B., D.W.B.), BCS-1459880 (R.B.B., D.W.B.), BCS-0850664 (R.B.B., D.W.B.), BCS-0314406 (R.B.B., D.W.B.). The authors gratefully acknowledge the support and creative practice of the Martu people, friends and family, who contributed their time, energy, and knowledge to help co-produce this work.

## Author contributions

R.B.B., D.W.B., P.M.V., D.T., C.M., and L.G. collected data; R.B.B. and C.M. collated and analyzed data; R.B.B., D.W.B., P.M.V., C.M., L.G., C.T.M., and T.M.W. wrote the paper.

## Competing interests
The authors declare no competing interests.
