## [Peer Review File · Nature Communications]

Seed dispersal by Martu peoples promotes the distribution of native plants in arid AustraliaReviewers' Comments:

Reviewer #1:

Remarks to the Author:

This study sought to investigate whether Martu people (an Indigenous group in northern Australia) have influenced the abundance and distribution of four traditional food plant species through seed dispersal and fire activity. The authors conducted plant surveys and ethnographic observations and used in an information theoretic approach (comparing generalised linear mixed models) to identify whether anthropogenic versus non-anthropogenic (soil, water) variables best explain the presence of the four study species. The authors concluded that two *Solanum* species showed stronger evidence of anthropogenic dispersal in the models compared with two edible, though less palatable species. Anthropogenic dispersal of non-crop plants is a poorly understood topic (particularly in Australia) and of great importance to Indigenous groups. It is also important for various fields in ecology, as until recently, the actions of ancient Indigenous populations have been a poorly considered factor in the biogeographic history of flora and fauna. It is noteworthy that the study investigates a 'mobile' human group, as the literature has primarily focused on sedentary or agricultural human groups. Though not highlighted by the authors, it is also notable to find stronger evidence for anthropogenic influence in more culturally significant species, compared with the less desired species, as it underscores the impact of culture on ecological processes.

I commend the authors for the creative way in which they've drawn together multiple datasets to investigate a topic that is difficult to study and quantify. It is a very interesting subject and great to see ecological and ethnographic data drawn together.

Validity/data/methodology

I found it hard to judge the analytical approach, as I got a little lost with all the datasets. In particular, the spatial and temporal scale of Martu activity covered is unclear. While the introduction discusses the evolutionary significance of human activity on plant species, the methods primarily refer to contemporary or very recent land-use activity by Martu people. My primary concern was the seeming use of recent presence/absence surveys to infer information about the impact of ancient activity (eg proximity to archaeological sites – or are these sites still used?). If it is not known how ancient 'pre-settlement'/pre-1960 activity is, this needs to be stated explicitly.

For instance, it's not clear whether the foraging patches spatially overlap or are directly related to the survey plots. A table that describes each dataset with a corresponding map would help. I was also wondering why the data were measured at different scales to the species survey plots? Having predictors at larger or smaller scale would make it difficult to quantify the relationship between the two.

Greater detail is needed on the tables and figures, particularly as readers see the results before methods. I couldn't find how the model weights were calculated and struggled to interpret Table 1. Does 'confirmed' under each model in Table 1 indicate that the prediction for each variable was correct? Additionally, a one sentence description of what information is to be obtained from each dataset could be stated in the methods to help readers interpret the data. For instance, was the goal of observing foraging activity was intended to identify the most frequently harvested species? The variable and model selection approach appears sound, though the AIC and other measures of fit were not reported. Did the best fit model for each species have the best score for each of the indices? The supplementary information was not attached so calculation of model performance couldn't be assessed.

Another concern is that all four species were generally absent from survey plots that had not received fire for > 3.5 years (presumably because they require fire for regeneration or to compete with other species). Doesn't this suggest that dispersal activity is insignificant compared with fire?

It is also hard to judge the overall impact of Martu people on the study species. The map in Figure 1 does not indicate the total size of the study area and how much of the species' distribution is captured. Including the distribution of the *Solanum* spp on the map would help readers judge the overall significance of the study area. I was also wondering if Martu communities exchange with other geographically distant people that would potentially result in longer range dispersal of *Solanum*. Is it

known whether other groups utilise the study species?

Clarity and context

Introduction –

could be improved with a statement that articulates that the extent to which Aboriginal Peoples in Aus moved plant species is unknown and debated. Although there is anecdotal ethnographic evidence summarised by Silcock (ref 12), the antiquity and scale of these practices are unknown. This creates greater justification for conducting surveys.

A sense of the anticipated results should be built on in the introduction. For instance, the first two paragraphs under the subheading 'Ecological modelling' in the Results (lines 136-160) contain information that provides context for the assumptions/methods in the study. This background information would be better earlier on in the article. The predictions for each variable stated in Table 1 need to be justified in the text before the results.

The last paragraph needs to be expanded on. Interpretation of the overall results would be greatly assisted with a summary of the specific hypotheses/relationships being tested with the surveys and models. For instance, does the model aim to understand the relative influence of soil/water vs anthropogenic variables on the presence and abundance of the study species in the study area? I would've liked more information on why the control species were selected aside from being a less desirable food source than Solanum. An ideal control species would occupy a similar distribution or share ecological traits to the study species (or better yet, one is promoted and the other is negatively impacted by fire). That way the impact of human activity will be easier to tease out.

The discussion –

would be strengthened if the focus was shifted away from domestication and expand on how humans provide ecosystem services – As an example. it appears that no animals move Solanum across the landscape (but is it easily wind dispersed?). It is mentioned in the introduction that the species are clonal – therefore anthropogenic dispersal may enhance the genetic diversity of the species than would otherwise occur through clonal reproduction. The significance of observing new pops in the years following dispersal should be emphasised, as it indicates that clonal propagation is not the only form of reproduction in the species. See more suggestions below.

Specific comments at lines:

23 - Abstract – good summary of background, methods and findings. Although doesn't mention that the surveys compare control vs study species, which is the more robust part of the methods. Could be improved with a significance statement.

39 – What is Yarnell's Camp Follower hypothesis?

45 – I suggest removing mention of endo and epizoochory as you first define them in the paragraph below.

47 – Important and relevant references:

Rossetto M, Ens EJ, Honings T, Wilson PD, Yap JS, Costello O, Round ER, Bowern C. From Songlines to genomes: Prehistoric assisted migration of a rain forest tree by Australian Aboriginal people. PLoS One. 2017 Nov 8;12(11):e0186663. doi: 10.1371/journal.pone.0186663.

B. Gott, Murnong - *Microseris scapigera*: A study of a staple food of Victorian Aborigines. Australian Aboriginal Studies, 2–18 (1983).

A. Lullfitz, M. Byrne, L. Knapp, S. D. Hopper, *Platysace* (Apiaceae) of south-western Australia: Silent story tellers of an ancient human landscape. Biological Journal of the Linnean Society (2020) <https://doi.org/10.1093/biolinnean/blaa035>.

53 – This introduction would be strengthened with some discussion of how the extent to which Aboriginal people dispersed useful species is debated or poorly understood.

Some references

P. Sutton, K. Walshe, *Farmers or Hunter-Gatherers? The Dark Emu Debate* (Melbourne University Press, 2021).

Silcock (2018) – already cited

B. Pascoe, Dark emu: black seeds: agriculture or accident? (Magabala Books, 2014).

55 – This paragraph nicely summarises how anthropogenic burning can influence plant distributions/abundances.

66 – Is Jukurr a Martu word?

68 – This sentence is good and should be reiterated through the rest of the manuscript. It would also appear that people influence the abundance not just the distribution of important food plants.

73 – You have reported a result in the intro. Just leave it at the citation, or cite this reference to present the question of whether increased anthropogenic fire would promote the two *Solanum* spp. It would be a suitable place to mention the different fire regenerative responses of the study species, creating a greater sense of the anticipated results for the reader.

74 – Were Martu people asked about *S. div* endozoochory?

What about wind dispersal? The seeds appear quite small in the photo of Fig 1.

75 - Please clarify meaning - the fruit is easy to obtain because few animals seem to consume it?

76 – suggest replacing “returned” with “transported”.

84 – suggest replacing “we draw on” with “we employ” or something similar as the way it is currently written sounds as though information is drawn from another study, not obtained in the current.

91 – It would be useful to describe some specific questions and expected results here. For instance the relative influence of anthropogenic versus non-anthropogenic factors. Are you testing for the abundance of the study plants, or just the presence/absence in each plot? And why were the control species selected – do they have other ecological similarities to the *Solanum* spp that would make one expect them to have the same distribution as the *Solanum*?

106 – state how many days harvesting activities were observed in total so that we can understand the proportion of days that *S. centrale* was harvested

117 – Are these foraging returns averaged per patch or is this the total return? And what defines a patch?

126 – please state how many years after the initial survey (readers haven’t looked at methods yet)

129 – it is unclear whether there are different camps at each year

142 – what is the source for the statement that a high density of fires correlates with intensity of human use over time? If in the methods, please refer the readers to the methods here.

145 – If prior to 1960, Martu primarily travelled via foot, is the prediction that vehicles have enabled greater post-settlement dispersal, or are people generally travelling/foraging less?

150 – what is load utility?

159 – Table 1 needs to be mentioned earlier in the manuscript. Following the info in this paragraph, is the prediction that *S. div* is only dispersed 2 km by Martu people? What about predictions for *S. centrale*?

187 – replace “dispersal” with “distribution”. The results suggest that recent burning is an important factor in the presence of *Solanum* spp. The discussion would be strengthened with discussion of the interaction between foraging (=dispersal) and burning (=recruitment/regen) in promoting *S. div*. Are there references you can cite to describe the relationship between fire and recruitment in *Solanum*?

193 – “Ethnographic observations of *S. div* harvesters also reveal substantial seed dispersal during harvest, transport, and processing that results in the plant successfully establishing new populations.” - Is this a quantified result or is the observation in Fig1 the only known example in the study? Wording should reflect that it is anecdotal evidence...

196 – So were surveys used to quantify abundance/density of the study species per plot, not just the presence?

199 – Given that all four species are generally absent from plots that haven’t received fire for >3.5 years, it seems that all species require frequent fire.

202 – Suggest changing opening sentence to “Our results suggest Martu people contribute to the widespread distribution of *S. centrale*, primarily through changes to the fire regime rather than active seed dispersal.”

204 – It would be nice to see some discussion of the evolutionary consequences of anthropogenic activity on the two species - i.e. increased clonal reproduction in *S. centrale* would yield different genetic impacts to the increased dispersal and genetic mixing that anthropogenic dispersal promotes in *S. div*.

205 – The manuscript would be much easier to follow if you mention the types of historic sites in the intro and set up expectations... and please clarify, is the expectation that S.cent would be closer to historic sites compared to S.div or in general?

Another question - is the expectation that anthropogenic dispersal contributes to the majority of the species' distribution in the study region or simply that it promotes it along anthropogenic pathways? What are the two types of historic sites?

206 – “but we expected this relationship to be weaker” - than with S.div?

207-211 – This description should go in the introduction of the species. Since S. cent is so easily dispersed, some discussion of how animal endozoochory may mask the impact of anthropogenic dispersal is warranted.

212 – As per my comment above, some more discussion of the impacts of anthropogenic versus natural dispersal would add to the significance of the findings. If I have understood Table 2 correctly, S. cent is present in fewer survey plots than S. div, yet it is apparently well dispersed by animals! However, need to link back to the statement in the introduction that no endozoochory has been observed for S. div.

234-236 – This is repeating the previous statement.

239 – It seems like the interaction between fire and dispersal was important for S.div presence.

253 – I would re-word, as both intentional and incidental seed dispersal by humans are worthy of investigation. And both processes are poorly understood in Australia.

258 – I wouldn't say the findings have much bearing on models of domestication. I suggest a more relevant point of discussion is 'human niche construction theory' as a framework to investigate human-mediated ecological change.

See references:

L. M. Amundsen-Meyer, Nature versus Culture: A Comparison of Blackfoot and Kayapó Resource Management. *Canadian journal of archaeology* 37, 219–247 (2013).

A. Lullfitz, J. Dortch, C. Hopper, Stephen D Pettersen, R. Reynolds, D. Guilfoyle, Human Niche Construction: Noongar Evidence in Pre-colonial Southwestern Australia. *Conservation and Society* 15, 201–216 (2017).

B. D. Smith, Niche construction and the behavioral context of plant and animal domestication. *Evolutionary Anthropology* (2007) <https://doi.org/10.1002/evan.20135>.

B. Smith, General patterns of niche construction and the management of “wild” plant and animal resources by small-scale pre-industrial societies. *Philosophical transactions of the Royal Society of London. Series B, Biological sciences* 366, 836–848 (2011).

K. N. Laland, M. J. O'Brien, Niche Construction Theory and Archaeology. *Journal of Archaeological Method and Theory* 17, 303–322 (2010).

270-278 – This is an interesting discussion and highlights that human-plant interactions are dynamic and culturally-variable. Can you relate it a little more to your results and to other studies in Aus?

For instance, casual seed broadcasting is reported in the following:

R. A. Hynes, A. K. Chase, Plants, Sites and Domiculture: Aboriginal Influence upon Plant Communities in Cape York Peninsula. *Archaeology in Oceania* 17, 38–50 (1982).

R. Jones, B. Meehan, “Plant foods of the Gidjingali: Ethnographic and archaeological perspectives from northern Australia on tuber and seed exploitation” in *Foraging and Farming: The Evolution of Plant Exploitation*, D. R. Harris, G. C. Hillman, Eds. (Unwin Hyman Ltd., 1989)

<https://doi.org/10.4324/9781315746425>.

R. Jones, “Ordering the landscape” in *Seeing the First Australians*, I. Donaldson, T. Donaldson, Eds. (George Allen & Unwin, 1985).

R. Cribb, Landscape as cultural artefact: Shell mounds and plants in Aurukun, Cape York Peninsula. *Australian Aboriginal Studies* 2, 60–73 (1998).

Kimber (1976) -already cited.

280 – Aren't Martu more 'semi-mobile'? They have semi-permanent settlement sites?

280-287 – Consider removing this paragraph (or at least merging some points with previous para). The last sentence is particularly overly speculative and the link with domestication is under-developed.

296 – Was *Eragrostis setifolia* traditionally part of the Martu diet? The citation refers to another Aboriginal group.

304 – Some information on the fire response of the control species would be great here.

310 – Information about the survey is not in Table 1...

311 – Why was transect orientation randomly selected? I'm wondering if there were well-established paths that would've been taken from camping places, and more likely for seed to have been dispersed along the way...

319 – were interviews conducted to obtain this knowledge? How many informants?

321 – citation is missing

339 – What is an ethnohistoric sites?

344 – I'm not convinced that winter fire density should be presented as a proxy for landscape use. Isn't the primary impact of burning that it affects the regeneration of the study species? Or is the expectation that more frequently used sites are more likely to be visited after *Solanum* harvest? Would this hold up in Winter (i.e. does *Solanum* fruit in Winter?)

347 – Did you confirm with Martu participants that hunting and fruit harvesting usually geographically correlated?

349 – Why is density summed and not averaged over the 30 year period?

361 – What is the 30 m plot? Why are the surveys all conducted at different scales? If there is a methodological reason, please cite. It would make interpretation easier if using consistent scales

365 – Where were existing fire history maps obtained? Please clarify this sentence – what are the categorical classifications and estimations that need to be converted?

387 – Slope, aspect, clay content are repeated below. Only need to list in the text once.

406 – I suggest merging the ethnographic methods section with the survey paragraph above the landscape variables. It could come under a 'survey methods' heading, with each survey type as a sub-heading (and justification for the survey). Otherwise readers will be confused about which survey is what.

411-413 – This should be in results section.

432 – What are the model weights and how are they calculated?

442 – Supplementary is not attached...

446 – Figure 1 – cm are missing from the scale for the *Solanum* fruit.

471 – Is the standardised coefficients from a correlation? Does global anthropogenic model mean that all anthropogenic models are included?

485 – Does "Plots present" mean the number of plots that the species was present in? It would be interesting to see the density of each species per plot as another variable.

Reviewer #2:

Remarks to the Author:

1. The paper is publishable with some revision as described below to ensure better contextualisation of the study setting especially for readers unfamiliar with Australian remote region Indigenous communities. There is some risk that readers will conclude Martu people are still deriving the major part of diet and lifestyle resources from the bush. If that is indeed the proposition from the authors, despite cultural and material changes over recent decades, then a more explicit account of this issue would be useful.

2. The scientific significance of the hypotheses and findings is robustly established in the subject area of human-plant interactions. The mixed methods of ecological transects, environmental data, fire history mapping and embedded ethnographic observation and interviewing are suitable to support the findings. The use of two control species to assess anthropogenic impacts on the distribution of the *Solanum* species presents a robust methodology. The big picture importance of addressing human/plant interactions beyond agriculture is presented to underline the intellectual significance of the paper.

3. Suggested revision to establish a better historical and cultural contextualisation of the study for readers:

- The authors speak to readers assuming unrealistic knowledge of traditional foraging as it has undergone changes in this remote Indigenous community setting. This issue may depend on how broadly the journal wishes to profile this kind of study. I am assuming that is one of the goals. If so, more specific wording will help us understand:

The extent to which the bush tomato has remained a significant part of diet, whether it has become more of a recreational food in light of dependence on purchased store foods and if so what has been the trajectory of this change. Are there data available about changing diets that can be cited more specifically? This issue is relevant to establishing the ongoing reliability of ethnographic observations over time (seemingly the last 20 years) in the bush and residential community settings. What has broad ethnographic research in this area told us about cultural change impacting hunting/foraging practices and beliefs?

What have been the impacts of motor vehicle usage on such cultural changes? Some topography would no longer be accessible compared to bush dwelling practices when desert life was still operating? So are the ethnographic observations telling us about what was once done in consuming the plant species or what are the practices now?

Dreaming stories are said to have certain environmental correlates, e.g. about what are preferred food sources for emu and bustard. Can the reader learn how or why such stories might indicate long observed behaviours of these species by Martu people? This may appear obvious for anthropologists committed to awareness of both clear and subtle cultural continuities. But sociocultural research in anthropology demonstrates that religious knowledge can derive as much from spiritual convictions as from everyday interactions with bush species. Is there an assumption about the significance of Dreaming stories on the part of the authors that can be clarified more specifically? Including the authors' approach to changes over time associated with sedentisation in residential communities where religious life has gone in new directions compared with bush living in earlier generations.

It would also be useful for readers to learn whether interviews about cultural practices and beliefs in earlier generations have been recorded, transcribed and analysed with cross-checking across different communities and among knowledgeable individuals. This to control for idiosyncratic versions of beliefs about species and spiritual entities not being adopted uncritically.

Some or all of the authors have completed ethnographic research with Martu people over the past 20 years. The reader would benefit from a succinct clarification as to how cultural changes in foraging behaviours have been addressed generally. What has been the role, if at all, of strategic traditionalism that may have resulted in self-conscious fashioning in usage of bush resources? Is the bush tomato a high-status bush food more so or less so now than in the past? As it stands there are a few snippets revealing knowledge the investigators may have of cultural changes, such as lines 327-8 noting the small number of elders with actual bush dwelling knowledge about camp sites and presumably associated foraging practices. The authors, though presumably not all of them, have carried out long-term community-based ethnographic research, and the reader would benefit hearing of the scope of those studies that provide the context for this particular paper.

- I suggest citing recent debates about absence / presence of agriculture in Australian Indigenous pre-colonisation modes of subsistence. Indeed, I don't see how the paper can be published without at least acknowledging this debate. Given the importance of the debate for contemporary Australian society there should be at least a note stating or acknowledging the authors' position on the works of Pascoe, Sutton & Walshe and other scholars where relevant. My reading of the article is that the authors take as a starting point a lack of what is normally regarded scientifically as agriculture across arid Australia in pre-European settlement times. However, readers will benefit from an explicit

comment on this issue.

David Trigger

Reviewer #3:

Remarks to the Author:

This work will be of significance to people in the field. There are many studies in archaeology/archaeobot/ethnobot where researchers have made observations about the co-occurrence of specific plants and sites of human occupation. However, I cannot think of any that have made attempts to provide quantitative data of this distribution along with qualitative observations and to provide control samples in doing so. For that reason alone, this is a paper that provides important new data for researchers. This work also makes use of data from a long-term study of a region of the Western Desert (Martu peoples) that further shows the value of such long-term research relationships.

I think that the paper is well produced with good supporting data and good quality illustrations and figures. The methods appear sound, showing positive or negative correlations in the presence/absence of key species against a combination of anthropogenic and non-anthropogenic variables. Throughout, there is an attempt to isolate variables or combinations of variables that are more strongly linked to human activity.

I have a few minor observations. Some of these may arise from my lack of familiarity in the statistical methods employed and the manner of data presentation. I am not familiar with the software employed for this study.

Spelling: line 422 of 'fiire'

On page 9 authors refer to five anthropogenic predictors, but then do not list them as they do 9 (nine?) most informative, non-collinear... I could identify four: distance to ethnohistoric site, water sources, site type, and winter fires. Perhaps list the anthropogenic predictors after a colon?

Water permanence. Line 317 states use of a permanence scale from 1 to 4, but these are summed in the statistics to 'water permanence'? Do the values of 1 to 4 have any role in the data presented?

Figure 1, line 444. Location of observed dispersal sites (brown). In my copy, I think these sites are marked with a red open circle with a red open cross.

Figure 2. combined variables on the Y-axis shown as fire density x water perm and site dist x fire density. What does the 'x' refer to? Are values being multiplied? In Figure 3. combination variables separated by a colon, e.g. dis site: Win fire. Is this a ratio variable? Needs clarification.

Figure 3. Non-specialists might appreciate a bit more text somewhere on the use of odds ratios and what is being displayed/shown with this data. This work will be of significance to people in the field. There are many studies in archaeology/archaeobot/ethnobot where researchers have made observations about the co-occurrence of specific plants and sites of human occupation. However, I cannot think of any that have made attempts to provide quantitative data on this distribution along with qualitative observations and to provide control samples in doing so. For that reason alone, this is a paper that provides important new data for researchers. This work also makes use of data from a long-term study of a region of the Western Desert (Martu peoples) that further shows the value of such long-term research relationships.

I think that the paper is well produced with good supporting data and good quality illustrations and figures. The methods appear sound, showing positive or negative correlations in the presence/absence

of key species against a combination of anthropogenic and non-anthropogenic variables. Throughout, there is an attempt to isolate variables or combinations of variables that are more strongly linked to human activity.

I have a few minor observations. Some of these may arise from my lack of familiarity in the statistical methods employed and the manner of data presentation. I am not familiar with the software employed for this study.

Spelling: line 422 of 'fiire'

On page 9 authors refer to five anthropogenic predictors, but then do not list them as they do 9 (nine?) most informative, non-collinear.... I could identify four: distance to ethnohistoric site, water sources, site type, and winter fires. Perhaps list the anthropogenic predictors after a colon?

Water permanence. Line 317 states use of a permanence scale from 1 to 4, but these are summed in the statistics to 'water permanence'? Do the values of 1 to 4 have any role in the data presented?

Figure 1, line 444. Location of observed dispersal sites (brown). In my copy, I think these sites are marked with a red open circle with a red open cross.

Figure 2. combined variables on the Y-axis shown as fire density x water perm and site dist x fire density. What does the 'x' refer to? Are values being multiplied? In Figure 3. combination variables separated by a colon, e.g. dis site: Win fire. Is this a ratio variable? Needs clarification. Elsewhere in the paper, *Solanum* sp. are grouped, but here they are separated by *Scaevola* sp in the key?

Figure 3. Non-specialists might appreciate a bit more text somewhere on the use of odds ratios and what is being displayed/shown with this data (e.g. text talks about correlations and relationships and uses the terms interchangeably, but these terms may mean different things). Also, maybe a statement here about the significance of a '1' value on the x-axis. Sometimes this is noted as a value on y-axis and sometimes it is omitted due to the scale of the axis but represented as a faint dotted line. Again, this could be clarified for non-specialists. It may also help the reader to more clearly indicate that that the top two species here are those likely dispersed by people and the lower two less-likely?

Table 1. Under column prediction, the term 'effect' is used as either positive or negative - but effect in what sense? As correlation? i.e. there is a positive correlation between site distance and presence (or frequency/ha?) of *Solanum* d? For *scaevola*, it is noted that model outcome Anth+fire for site distance is 'farther from sites'. But what is meant here - is this qualitative or quantitative? Some explanatory text could be included with this table to help the reader navigate this summary data presentation.

Point by point response

Reviewer #1 (Remarks to the Author):

This study sought to investigate whether Martu people (an Indigenous group in northern Australia) have influenced the abundance and distribution of four traditional food plant species through seed dispersal and fire activity. The authors conducted plant surveys and ethnographic observations and used in an information theoretic approach (comparing generalised linear mixed models) to identify whether anthropogenic versus non-anthropogenic (soil, water) variables best explain the presence of the four study species. The authors concluded that two *Solanum* species showed stronger evidence of anthropogenic dispersal in the models compared with two edible, though less palatable species.

Anthropogenic dispersal of non-crop plants is a poorly understood topic (particularly in Australia) and of great importance to Indigenous groups. It is also important for various fields in ecology, as until recently, the actions of ancient Indigenous populations have been a poorly considered factor in the biogeographic history of flora and fauna. It is noteworthy that the study investigates a 'mobile' human group, as the literature has primarily focused on sedentary or agricultural human groups. Though not highlighted by the authors, it is also notable to find stronger evidence for anthropogenic influence in more culturally significant species, compared with the less desired species, as it underscores the impact of culture on ecological processes.

I commend the authors for the creative way in which they've drawn together multiple datasets to investigate a topic that is difficult to study and quantify. It is a very interesting subject and great to see ecological and ethnographic data drawn together.

Validity/data/methodology

I found it hard to judge the analytical approach, as I got a little lost with all the datasets. In particular, the spatial and temporal scale of Martu activity covered is unclear. While the introduction discusses the evolutionary significance of human activity on plant species, the methods primarily refer to contemporary or very recent land-use activity by Martu people. My primary concern was the seeming use of recent presence/absence surveys to infer information about the impact of ancient activity (eg proximity to archaeological sites – or are these sites still used?). If it is not known how ancient 'pre-settlement'/pre-1960 activity is, this needs to be stated explicitly.

Yes, we acknowledge that our proxies for present and past land use required further discussion and clarity. We believe we achieved this in the redrafted manuscript at in the introduction 35-69, We have now emphasised how the research design is geared toward investigating whether plant abundance and presence from contemporary ecological surveys (transect surveys) shows evidence of a continuous history of land use from past to present, past in the form of distance to former habitation sites, present in the form of the intensity of land use as observed today. Because past and present proxies of use involve

different land use patterns, we include both types of proxies. It could also be the case that contemporary plant distributions are affected only by past land use, or only by present land use and not past use. Our analysis clearly shows both past and present land use influences dispersal. We also have included more discussion of other studies that have used distance to archaeological sites as proxies for past land use. We have signposted this in the introduction, the outline of hypotheses, discussion and methods, While Martu mobility is different today than before European contact, they retain very high residential and logistical mobilities, which we now highlight in the first paragraph of the Results section. We have published extensively on this very theme in a range of international journals for 20 years e.g.(e.g. Bliege-Bird et al. 2008).

As explained more clearly now, our focus is on how long-term legacies of land use have shaped the contemporary distribution of plants. Therefore one of the aims of the study was to investigate whether the location of contemporary patches of key plants, especially *S. diversiflorum*, was correlated with the distribution of ethnohistoric and archaeological sites. The same approach has been used in several important studies (now more clearly cited); these have used ecological surveys to investigate whether specific food plants are more abundant in proximity to archaeological sites, some of which have indeed been abandoned for 2000 years or more (e.g. Pavlik et al. 2021). With respect to the specific question of how recent presence/absence surveys can be used to interrogate correlations with pre-Contact ethnohistoric settlement and mobility, this is an empirical question. If the seedbank persists over time, even if people are no longer living at the site, the patches will still be more likely to be present. Furthermore, people still revisit many of these sites, such that past and present land use overlaps in some spatial locales. We have thus introduced further contextual data throughout the paper. For example, at lines lines 81 – 81: “Many of the archaeological sites recorded by PV with Martu custodians in the 1980s had started to be re-occupied with the advent of the outstation movement. Both the pre-contact assemblages and contemporary plant use within the site locale, were recorded (Veth 1993; 2006; Walsh 1987). ,

To better clarify this research strategy, we now establish the rationale for an introductory section titled “Martu: foraging and mobility”, before we discuss specifics regarding methods used in data collection. New text is introduced from lines 70 - 114:

This work is part of a long-term collaborative ethnographic, ecological, and ethnoarchaeological project with Martu communities located in the heart of their homelands. Martu and Kulyakartu Native Titles encompass more than 150,000 sq km of the Great and Little Sandy Deserts ecoregions, and our study area comprises a subset region of about 42,000 sq km (Figure 1). While intermittent interaction between some Martu and Europeans began early in the 20th century, many avoided contact with settler-colonial invasion until the 1960s. Many remote-living Martu were first contacted in the 1960s during government welfare patrols tasked with evaluating the drop zone for inter-ballistic missiles that were eventually launched in the International Weapons Research Establishment, Australia's principal contribution to Cold War efforts. The last groups left the desert in 1967, but reoccupation of Martu lands began in the 1980's

with the establishment of three remote communities at Punmu, Parnngurr, and Kunawarrtji. One of the authors who is Martu (DT) was born in Martu country before the hiatus and was involved in all aspects of the homelands movement, having multi-decadal experience in language translation; while another (PV) began archaeological programs in collaboration with remote-living Martu families in the mid-1980s. Many of the archaeological sites recorded by PV with Martu custodians in the 1980s had started to be re-occupied with the advent of the outstation movement. Both re-contact archaeological assemblages and contemporary plant use around the sites, were recorded with ethnobotanists (Veth 1993; 2006; Walsh 1987). RBB and DWB began work here in 2000 as Martu were preparing their Native Title claim, which was determined in 2002.

On return to their homelands, Martu reinvigorated their customary subsistence and landscape burning practices^{22, 30,31}, and today many remote-living families continue to maintain fundamental aspects of their foraging livelihoods, typically spending about 25% of their days in hunting or gathering activities^{55, AND Bliege Bird et al. 2016 SAR}. In a series of articles, we report quantitative details of foraging time allocation, labor decisions, bush food yields, return rates from regular hunting and gathering practices, the role of cultural burning in facilitating hunting strategies, the ecological role of these practices, and significant changes in subsistence over the last half-century^{24,42-44,46,47,51-59,61}.

While the three remote communities in the Martu Native Title lands have semi-permanent infrastructure – including in each some housing, a primary school, a small shop, and a health clinic – it would be misleading to describe Martu as “settled” or even “semi-settled”. Most exercise extremely high mobility, shifting residence many times over the course of a year across the remote communities and throughout settlements and towns beyond the Native Title boundaries. The composition of Martu residential communities and foraging groups is thus constantly fluctuating⁵⁵. Their mobility and foraging activities are vital in maintaining the landesque capital of their estates, fundamentally entangled in a web of both social and ecological interactions between people, fire, plants, and animals that make the landscape of country “ngurra-ra”, or homeland⁵⁶. These interactions are material manifestations of the creative epoch (The Dreaming), in which people re-enact the process of creation through the maintenance and support of ecological networks^{43,44,50,57,58}.

In the years of dialogue with Martu while foraging, adults have often acknowledged their role in the propagation of useful plants, as has also been reported by others²². People offer various explanations for why some plants grow so densely in some places and not others by noting that this was where people used to camp and clean seed, or that a particular locale had been popular for fruit picking “for a long time”. Continued use of the region is thought to have supported the growth of more plants. Alternatively, Martu have noted that fire keeps some plants coming back, and that a fire burned in winter produces more fruit. Many describe how as youngsters they were told to clean bush tomatoes within the same place (patch), so that the plants and their fruit/seeds would always be there. This does not translate to management of plants, *per se*, in the way that farmers might: they are acquiring and sharing resources with others as a meaningful means of building relational and ritual wealth^{50,56,59}. In so doing, this process of making food good for consumption creates and shapes landscapes, which are both the source of food and socio-religious capital. It is this precise knowledge that has ritual significance: the knowledge that Martu belong to the country and to the plants and animals that live within it, and they have both a spiritual *and* material role to play in making that country productive.

For instance, it’s not clear whether the foraging patches spatially overlap or are directly related to the survey plots. A table that describes each dataset with a corresponding map would help. I was also wondering why the data were measured at different scales to the species survey plots? Having predictors at larger or smaller scale would make it difficult to quantify the relationship between the two.

Table 2 now summarizes the transect surveys, and Figure 1 shows the locations of all transects and dinner camps around which we conducted dispersal survey.

As we now detail in the analysis, our transect survey counts were deliberately aggregated to match the scale of the imagery with the predictive layers derived from remote sensing. The dispersal site surveys are a separate analysis and we did not use these as remote sensing predictors. As such, their slightly different scale (50 meters vs 30 meters) does not affect the results.

Greater detail is needed on the tables and figures, particularly as readers see the results before methods. I couldn't find how the model weights were calculated and struggled to interpret Table 1. Does 'confirmed' under each model in Table 1 indicate that the prediction for each variable was correct? Additionally, a one sentence description of what information is to be obtained from each dataset could be stated in the methods to help readers interpret the data. For instance, was the goal of observing foraging activity was intended to identify the most frequently harvested species?

Agreed with reviewer comments and we have remedied this issue We have completely re-configured Table 1 and the figure captions now provide the detail requested by the reviewer.

The variable and model selection approach appears sound, though the AIC and other measures of fit were not reported. Did the best fit model for each species have the best score for each of the indices? The supplementary information was not attached so calculation of model performance couldn't be assessed.

Agreed and has been addressed with a full description of the top ten statical models for predicting the presence and absence of key species)Supplementary Table 1)

Another concern is that all four species were generally absent from survey plots that had not received fire for > 3.5 years (presumably because they require fire for regeneration or to compete with other species). Doesn't this suggest that dispersal activity is insignificant compared with fire?

In both the results and discussion sections we clarify that the patterns of key species distributions reflect both a legacy of anthropogenic fires and dispersal resulting from *in situ* food processing. We believe we avoid a teleology by showing co-variance against known and previously occupied sites. For example, at lines 410 – 414 we note “Ethnographic observations of *S. diversiflorum* harvesters, and surveys of food consumption sites, reveal substantial seed dispersal during harvest, transport, and processing that results in the plant successfully establishing new populations. As predicted in this study, plant abundance and presence shows evidence of both contemporary and past landscape use, being more likely to be present and at higher density in areas heavily used for contemporary hunting activities that are closer to minor occupation sites and near permanent water sources”.

It is also hard to judge the overall impact of Martu people on the study species. The map in Figure 1 does not indicate the total size of the study area and how much of the species' distribution is captured. Including the distribution of the *Solanum* spp on the

map would help readers judge the overall significance of the study area. I was also wondering if Martu communities exchange with other geographically distant people that would potentially result in longer range dispersal of *Solanum*. Is it known whether other groups utilise the study species?

These are useful comments from the reviewer which we have addressed. We have added a distribution map in Figure 1. which shows the size of the study area (42,000 km²). Inset C in Figure 1 now shows the known distribution of all four species.

There is also a request for more detail on Aboriginal use of these plants more broadly.. We also included more relevant detail on harvesting and dispersal, in the introduction and with the majority of new text in the results section. For example , with respect to the transport of *Solanum* we note at lines 195 – 197 “In maintaining their especially high residential mobility, Martu will also sometimes transport a load (up to 10kg) of pre-processed *S. diversiflorum* to share with kin in other communities, occasionally even beyond their Native Title boundaries.”

Clarity and context

Introduction –

could be improved with a statement that articulates that the extent to which Aboriginal Peoples in Aus moved plant species is unknown and debated. Although there is anecdotal ethnographic evidence summarised by Silcock (ref 12), the antiquity and scale of these practices are unknown. This creates greater justification for conducting surveys.

A sense of the anticipated results should be built on in the introduction. For instance, the first two paragraphs under the subheading ‘Ecological modelling’ in the Results (lines 136-160) contain information that provides context for the assumptions/methods in the study. This background information would be better earlier on in the article. The predictions for each variable stated in Table 1 need to be justified in the text before the results.

We agree and have introduced new text in the reworked introduction about addressing knowledge gaps in the agency of humans in species distributions and “ sense of the anticipated results”. We have aimed to provide more justification for why the surveys were conducted, without creating redundant text which is outlined in detail in the Methods section. At lines 467- 480 we summarise the current (*Dark Emu*) debate about seed dispersal being cast as *either* farming or foraging; concluding that intentionality behind seed dispersal does not necessarily influence the ecological consequences. Clearly, we conclude that the combined influence of processing propagules and local firing practices have established a positive feedback loop which covaries with both pre- and post-contact site use.

The last paragraph needs to be expanded on. Interpretation of the overall results would be greatly assisted with a summary of the specific hypotheses/relationships being tested with the surveys and models.

We have aimed to clarify these points in the recrafted concluding paragraph to the Introduction, which now reads (lines 62 – 68):

“Here, we ask whether a substantial history of hunting, gathering, seed dispersal, burning and other landscape use by Aboriginal people (Martu) has influenced the distribution of four edible plant species common to the arid deserts of Central and Western Australia, which include two bush tomatoes (*Solanum diversiflorum* and *S. centrale*), a seed grass (*Eragrostis spp.*), and a small forb (*Scaevola parvifolia*). Our approach builds on earlier work in four complementary ways: 1) we employ a hypothetico-deductive methodology with a controlled comparison research design to 2) test potential causal mechanisms for plant dispersal, with 3) quantitative data on plant distributions, on a landscape with 4) a long history (ca 50 ka years) of highly mobile hunting and gathering practices”.

I would've liked more information on why the control species were selected aside from being a less desirable food source than *Solanum*. An ideal control species would occupy a similar distribution or share ecological traits to the study species (or better yet, one is promoted and the other is negatively impacted by fire). That way the impact of human activity will be easier to tease out.

This is a salient point from the reviewer which we have now addressed in several places in both the Introduction and in Inset C from Figure 1.

The discussion –

would be strengthened if the focus was shifted away from domestication and expand on how humans provide ecosystem services – As an example. it appears that no animals move *Solanum* across the landscape (but is it easily wind dispersed?). It is mentioned in the introduction that the species are clonal – therefore anthropogenic dispersal may enhance the genetic diversity of the species than would otherwise occur through clonal reproduction. The significance of observing new pops in the years following dispersal should be emphasised, as it indicates that clonal propagation is not the only form of reproduction in the species. See more suggestions below.

In the new section titled “The plants: mechanisms of dispersal” we now detail known and possible agents of dispersal for each plant spp. We don't think it is relevant to note that seeds are not wind dispersed, as we describe that they are held within the fruit and must be removed before consumption. The relative large seeds of large fruits are not wind dispersed. and there are no studies conducted in arid Australia that record significant non-human consumption of the fruit. Martu insist that people are the only significant agents of *diversiflorum* dispersal, given that only people can pick, transport, process, and consume the fruit. The plant is exceptionally thorny and the fruit has a thorny calyx attached to the thorny stems. The fruit the fruit has to be picked from underneath with highly dexterous actions. On rare occasions Martu have pointed out *diversiflorum* fruit that The Hill Kangaroo have nibbled, however this sole example does so in a way that leaves the seeds attached to the calyx on the stem. *Diversiflorum* is not clonal - only *S. centrale* is clonal and evidence for human dispersal is weak, as predicted.

Specific comments at lines:

23 - Abstract – good summary of background, methods and findings. Although doesn't mention that the surveys compare control vs study species, which is the more robust part of the methods. Could be improved with a significance statement.

We agree and have added a significance statement.

39 – What is Yarnell's Camp Follower hypothesis?

Agree: we now explained Yarnell's hypothesis.

45 – I suggest removing mention of endo and epizoochory as you first define them in the paragraph below.

Agree and removed.

53 – This introduction would be strengthened with some discussion of how the extent to which Aboriginal people dispersed useful species is debated or poorly understood.

We have now re-written the Introduction and emphasised this point.

55 – This paragraph nicely summarises how anthropogenic burning can influence plant distributions/abundances.

66 – Is Jukurr a Martu word?

Yes. We have substituted this with the common term used for Aboriginal creation narratives: "The Dreaming".

68 – This sentence is good and should be reiterated through the rest of the manuscript. It would also appear that people influence the abundance not just the distribution of important food plants.

Agreed and this been done.

73 – You have reported a result in the intro. Just leave it at the citation, or cite this reference to present the question of whether increased anthropogenic fire would promote the two *Solanum* spp. It would be a suitable place to mention the different fire regenerative responses of the study species, creating a greater sense of the anticipated results for the reader.

Agreed. We now detail that each of the study species is an early- to mid-successional plant, and without low intensity burning they are crowded out of the landscape by the encroaching climax species (i.e. spinifex grass). The successional stages and many

analyses of Martu fire regimes are detailed in a series of publications that we now cite throughout the paper.

74 – Were Martu people asked about *S. div* endozoochory?

We now specifically address this issue throughout the paper. *S. diversiflorum* seeds are never eaten – they are exceptionally bitter and have alkaloids, as does the membrane-like tissue in which they are encased, which always badly burns your mouth. Martu (and as we now note, kangaroos) always avoid consuming the seeds – the fruit must always be processed. *S. centrale* is a different matter. In general, our Martu collaborators have not entertained human endozoochory – indeed they are reluctant to discuss such a human digestive function of a fruit. Our senior Martu co-author eschews speculation on possible ways that anthropogenic endozoochory may influence the dispersal of these plants. The results indicate that whatever the mechanisms may be, people are not only weakly implicated in the dispersal of this species.

What about wind dispersal? The seeds appear quite small in the photo of Fig 1.

The seeds are not small, they are the size of a capsicum seed, thus unlikely to be wind dispersed. See above, and our new section on the mechanisms of dispersal for each plant, which appears at lines 159 - 244.

75 - Please clarify meaning - the fruit is easy to obtain because few animals seem to consume it?

No, we meant to say that the fruit is high ranked, not easy to obtain, and that people are likely to be a significant agent of dispersal because few other animals seem to consume the fruit. We have now clarified this point in a new section on mechanisms of dispersal. In Martu country, especially *S. diversiflorum* and *S. centrale*, have very high return rates (yield per unit time spent harvesting and processing). From an energetics sense, and also in the seasonal and geographical predictability of patches, they are relatively easy to obtain (that is they do not require major “search time” as such). But as we outline in detail now in the section on “mechanism of dispersal”, *S. diversiflorum* has a suite of adaptations that makes accessing and harvesting them challenging; traits that preclude most other animals from consuming the fruit.

76 – suggest replacing “returned” with “transported”.

Agreed and done.

84 – suggest replacing “we draw on” with “we employ” or something similar as the way it is currently written sounds as though information is drawn from another study, not obtained in the current.

Agreed. We make this change throughout the manuscript.

91 – It would be useful to describe some specific questions and expected results here. For instance the relative influence of anthropogenic versus non-anthropogenic factors. Are you testing for the abundance of the study plants, or just the presence/absence in each plot? And why were the control species selected – do they have other ecological similarities to the Solanum spp that would make one expect them to have the same distribution as the Solanum?

This point has been systematically addressed in our responses to the reviewer (see above). We have aimed to make these questions/matters much clearer now in the Results section and in the preceding (sub)sections.

106 – state how many days harvesting activities were observed in total so that we can understand the proportion of days that *S. centrale* was harvested

The number of days harvesting activities observed in surveys were and are explicitly discussed in the in the “Background and ethnographic methods” and in the “Results” sections.

117 – Are these foraging returns averaged per patch or is this the total return? And what defines a patch?

Agree this should be clarified. We have provided more detail and context in the “Results”: in-patch foraging return rates are reported as kcal/hr averaged across foraging bouts. This expresses the total edible yield harvested by a single forager on a given bout of collecting and processing within a given patch.

126 – please state how many years after the initial survey (readers haven’t looked at methods yet)

We have added this.

129 – it is unclear whether there are different camps at each year

We have aimed to make this clearer throughout the paper. There are many camps used throughout each year. As we emphasize, Martu retain very high logistical and residential mobility, despite the fact that their three remote communities have more “permanent” infrastructure. These now serve as base residential camps for highly mobile people.

142 – what is the source for the statement that a high density of fires correlates with intensity of human use over time? If in the methods, please the readers to the methods here.

We have made this clearer throughout the manuscript. The relationship between density of fires and intensity of site has been the focus of much of our previous work with Martu, which is now cited throughout.

145 – If prior to 1960, Martu primarily travelled via foot, is the prediction that vehicles have enabled greater post-settlement dispersal, or are people generally travelling/foraging less?

No, we do not test any hypotheses related to the use of motor vehicles in this publication, nor are we testing any hypotheses related to traveling or foraging less often. We give more detail now about how there are some differences between pre-contact mobility and mobility today. But overall, we focus on whether we can detect any effect of anthropogenic dispersal, regardless of changes in transport technology or how much people are actually foraging or where they are going. I hope this helps clarify the issue. We have explicitly addressed this issue previously, including the paper: For example, see Zeanah, D.W., Coddling, B., Bird, D.W., Bliege Bird, R., and P. Veth 2015 Diesel and damper: Changes in seed use and mobility patterns following contact amongst the Martu of Western Australia. *Journal of Anthropological Archaeology* 39: 51-62.

150 – what is load utility?

Load utility is the proportion of load that is comprised of useful parts. We now clarify this in the “Hypotheses and predictions” section.

159 – Table 1 needs to be mentioned earlier in the manuscript. Following the info in this paragraph, is the prediction that S.div is only dispersed 2 km by Martu people? What about predictions for S. centrale?

We have not yet calculated the predicted field processing thresholds for either S. div or S. centrale. This is the focus of analyses currently underway, so this statement is based on numerous conversations with Martu and our own qualitative field observations. For example, the S.div patch is further than about 2 km from a residential, foraging camp, or transport vehicle, the harvested load will be processed at or very near the patch. S. centrale is never field processed because the utility of a load of unprocessed S. centrale is close to 100% (meaning its is all edible).

187 – replace “dispersal” with “distribution”. The results suggest that recent burning is an important factor in the presence of Solanum spp. The discussion would be strengthened with discussion of the interaction between foraging (=dispersal) and burning (=recruitment/regen) in promoting S.div. Are there references you can cite to describe the relationship between fire and recruitment in Solanum?

We agree with the reviewer and think this is an excellent question. In several decades of researching the effects of anthropogenic burning and pyrodiversity on landscape

dynamics, we have found very few studies that systematically investigate the relationship between variability in fire interval, fire intensity/magnitude/size and resultant vegetative succession relative to recruitment of *Solanum* bush tomatoes. There is one study we now report on in the Discussion showing *S. centrale* presence is affected by fire frequency, likely through stimulating clonal reproduction (Pattison et al. 2019).

193 – “Ethnographic observations of *S. div* harvesters also reveal substantial seed dispersal during harvest, transport, and processing that results in the plant successfully establishing new populations.” - Is this a quantified result or is the observation in Fig1 the only known example in the study? Wording should reflect that it is anecdotal evidence...

Agreed. This is not anecdotal; it is the result of the long-term dispersal site survey. We clarify this now in the re-write of the Results section.

196 – So were surveys used to quantify abundance/density of the study species per plot, not just the presence?

Yes, but the models were not particularly informative. We have included them now though on the request of the reviewer.

199 – Given that all four species are generally absent from plots that haven't received for for >3.5 years, it seems that all species require frequent fire.

Yes which is why these species were chosen for comparison, but we note that summer lightning causes fire also - not just people [see Bliege Bird et al. 2008, 2012, 2016, 2018 cited in the manuscript for full details].

202 – Suggest changing opening sentence to “Our results suggest Martu people contribute to the widespread distribution of of *S. centrale*, primarily through changes to the fire regime rather than active seed dispersal.”

Agreed and this is a good suggestion. This has now introduced into the re-write.

204 – It would be nice to see some discussion of the evolutionary consequences of anthropogenic activity on the two species - i.e. increased clonal reproduction in *S. cent* would yield different genetic impacts to the increased dispersal and genetic mixing that anthropogenic dispersal promotes in *S. div*.

We are currently working on this issue with colleagues in a large genetic study of *S. div* from hundreds of patches across the Martu homelands. The expected comparisons between *S. div* and *S. centrale* will require significantly more work before we can make predictions or speculate here. We see that as a stand-alone study some 5 years from completion. It should be an excellent companion and development study of the foundational research reported on here.

205 – The manuscript would be much easier to follow if you mention the types of historic sites in the intro and set up expectations... and please clarify, is the expectation that *S.cent* would be closer to historic sites compared to *S.div* or in general?

We have revised the manuscript to make this much clearer both in the Introduction and especially in the section “Background and Ethnographic methods”.

Another question - is the expectation that anthropogenic dispersal contributes to the majority of the species' distribution in the study region or simply that it promotes it along anthropogenic pathways?

Agreed . We have spent significant time in the explication of the hypotheses in the revised manuscript; specifically we expect people to be one influence on plant distribution, and to differing degrees for each species.

What are the two types of historic sites?

We are now more specific in our definitions of sites and consistent throughout in our terminology. Both ethnohistoric and archaeological sites are further categorized as major and minor as we more fully describe them in the text at various places.

206 – “but we expected this relationship to be weaker” - than with *S.div*?

Agreed. We have modified the text to now read: “...although the relationship seems less straightforward than in the case of *S. diversiflorum*.”

207-211 – This description should go in the introduction of the species. Since *S. cent* is so easily dispersed, some discussion of how animal endozoochory may mask the impact of anthropogenic dispersal is warranted.

Agreed. As noted above, we now have many more details on this theme throughout, and especially in the “mechanisms of dispersal” section.

212 – As per my comment above, some more discussion of the impacts of anthropogenic versus natural dispersal would add to the significance of the findings. If I have understood Table 2 correctly, *S. cent* is present in fewer survey plots than *S. div*, yet it is apparently well dispersed by animals! However, need to link back to the statement in the introduction that no endozoochory has been observed for *S. div*.

Agreed. As noted above, we now have many more details on this theme throughout, and especially in the “mechanisms of dispersal” section.

234-236 – This is repeating the previous statement.

Agreed and removed.

239 – It seems like the interaction between fire and dispersal was important for S.div presence.

We now clarify the discussion of the model results which show that fire is *not* significant when dispersal covariates are added to the model.

253 – I would re-word, as both intentional and incidental seed dispersal by humans are worthy of investigation. And both processes are poorly understood in Australia.

This is a good point. We significantly re-worked this theme in the new Discussion.

258 – I wouldn't say the findings have much bearing on models of domestication. I suggest a more relevant point of discussion is 'human niche construction theory' as a framework to investigate human-mediated ecological change.

We believe the redrafted Discussion addresses this theme, however do not think a new suite of hypotheses more explicitly derived from niche construction theory are warranted in this paper. It would substantively change the direction of the paper and the relevance of many of the methods relied on. However, we do now refer more explicitly to our own work with Martu on human niche construction, especially Bliege Bird et al 2013 and Bird et al 2016.

270-278 – This is an interesting discussion and highlights that human-plant interactions are dynamic and culturally-variable. Can you relate it a little more to your results and to other studies in Aus?

Agreed. This was also suggested by another reviewer, and we have added a new discussion of the Pascoe *Dark Emu* debate with critiques including one by co-author PV

280 – Aren't Martu more 'semi-mobile'? They have semi-permanent settlement sites?

We believe this theme his should be much clearer throughout the paper now. We detail (and cite many of our previous publications that demonstrate) that it is unwarranted to describe Martu as "semi-mobile" in their large foraging territories and numerous minor and major remote site locales; although we explicitly note now have three community bases in their Native Title. These can vary in populations ranging from 20 – well over 500 people depending on ceremonial, resource and funerary cycles. We now discuss mobilities more clearly in the "mechanisms of dispersal" section, with more elaboration in both the Results and Methods section.

280-287 – Consider removing this paragraph (or at least merging some points with previous para). The last sentence is particularly overly speculative and the link with domestication is under-developed.

Agreed and we have modified the text.

296 – Was *Eragrostis setifolia* traditionally part of the Martu diet? The citation refers to another Aboriginal group.

Yes, and we clarify this traditional dependence in the text now.

304 – Some information on the fire response of the control species would be great here.

Agreed. We have aimed to do this wherever possible. Overall, however, there is very little information available other than our analyses demonstrating preference for more recently burnt patches (early & mid-succession)

310 – Information about the survey is not in Table 1...

This has been rectified with information now in the new tables and provision of more detailed captions.

311 – Why was transect orientation randomly selected? I'm wondering if there were well-established paths that would've been taken from camping places, and more likely for seed to have been dispersed along the way...

We think this is an intriguing view; however as now expanded on, Martu mobility doesn't operate as establish pathways in the desert: paths are much more probabilistic and configured more like corridors.

319 – were interviews conducted to obtain this knowledge? How many informants?

As we detail now in the manuscript, our long-term project includes extensive participatory action research and dialogues over many hundreds of trips (as per Martu insistence) with free, prior and informed consent at all times. The ethnographic work is longitudinal, with full participation in subsistence practices and land use, and with quantitative recording of all activities participated in. We didn't intend this statement in the draft to appear as a hypothesis test, it was for context only.

321 – citation is missing

This has now been added in the re-write.

339 – What is an ethnohistoric sites?

See above. We are now more specific in our definitions of site types and more consistent throughout the paper in our terminology. We now clearly define what we term ethnohistoric vs. archaeological sites in the 2nd para of the Methods section.

344 – I'm not convinced that winter fire density should be presented as a proxy for

landscape use. Isn't the primary impact of burning that it affects the regeneration of the study species?

Or is the expectation that more frequently used sites are more likely to be visited after Solanum harvest? Would this hold up in Winter (i.e. does Solanum fruit in Winter?)

This is an important point and we thank the reviewer for this comment. We now draw the readers' attention to our well-published development of this theme of fire being a good proxy of landscape use, as reported in Bliege Bird et al. 2008. Fire density does not correlate well at all with measures of the fire regime, such as time since fire Shannon diversity.

347 – Did you confirm with Martu participants that hunting and fruit harvesting usually geographically correlated?

This is based on our 20+ years of gathering and hunting with Martu, and quantitatively recorded foraging trips and individual focal follows. When fruit harvesting occurs, it is almost always in the context of logistical forays out from the community, in which the party leaves together, establishes a temporary foraging camp in the region of the day's activities, and while some are hunting, others may collect Solanum. We detail this now in the 4th para of the Methods section.

349 – Why is density summed and not averaged over the 30 year period?

This is a good point. We use this because landscape use builds on the past use in a summative way. With the pre-Contact archaeology this often presents as palimpsests of occupation through space and over time but tethered to locales with more permanent water and floristic diversity.

361 – What is the 30 m plot? Why are the surveys all conducted at different scales? If there is a methodological reason, please cite. It would make interpretation easier if using consistent scales

This issue has now been clarified (see our comments about scale above).

365 – Where were existing fire history maps obtained? Please clarify this sentence – what are the categorical classifications and estimations that need to be converted?

These fire history maps were generated by our team in a series of publications now well cited throughout the updated manuscript. The methods for map construction are cited now in the text and represent many hundreds of hours of work

387 – Slope, aspect, clay content are repeated below. Only need to list in the text once.

406 – I suggest merging the ethnographic methods section with the survey paragraph above the landscape variables. It could come under a 'survey methods' heading, with each survey type as a sub-heading (and justification for the survey). Otherwise readers will be confused about which survey is what.

Agreed. This has now been re-organized in the re-write.

411-413 – This should be in results section.

432 – What are the model weights and how are they calculated?

Agreed. This has been clarified and is now in the supplementary table.

442 – Supplementary is not attached...

We were unsure about why the supplementary materials were not attached from the publisher.

446 – Figure 1 – cm are missing from the scale for the Solanum fruit.

This has been rectified.

471 – Is the standardised coefficients from a correlation? Does global anthropogenic model mean that all anthropogenic models are included?

485 – Does “Plots present” mean the number of plots that the species was present in? It would be interesting to see the density of each species per plot as another variable.

We have clarified both points in the redrafting of the manuscript.

Reviewer #2 (Remarks to the Author):

1. The paper is publishable with some revision as described below to ensure better contextualisation of the study setting especially for readers unfamiliar with Australian remote region Indigenous communities. There is some risk that readers will conclude Martu people are still deriving the major part of diet and lifestyle resources from the bush. If that is indeed the proposition from the authors, despite cultural and material changes over recent decades, then a more explicit account of this issue would be useful.

This is an excellent point. We have re-written the manuscript, which now includes base-line details throughout (but especially in the new section titled “Martu: foraging and mobility”) on contemporary subsistence for remote-living Martu. We direct the reader now to our previous publications with analyses of quantitative records of time allocation, yields, food sharing from a wide foraging activities and land/resource use, including discussions of changes in subsistence from past to present. Our previous analyses consistently estimate total bush food gathering + hunting effort as around 25% of days, which is comparable to estimations made by other anthropologists/ethnographers/ethnobotanists who have worked in remote communities. For this paper, we emphasise that people still do a lot of plant harvesting and have been

doing so over the last 30 years with return to their homelands. What is especially important for this analysis is that both past and present landscape use affects current distributions of some plants. For *Solanum* fruit especially, if contemporary harvesting were not occurring, we would not expect that current landscape use would predict distribution and abundance, only past landscape use. This is clearly not the case.

2. The scientific significance of the hypotheses and findings is robustly established in the subject area of human-plant interactions. The mixed methods of ecological transects, environmental data, fire history mapping and embedded ethnographic observation and interviewing are suitable to support the findings. The use of two control species to assess anthropogenic impacts on the distribution of the *Solanum* species presents a robust methodology. The big picture importance of addressing human/plant interactions beyond agriculture is presented to underline the intellectual significance of the paper.

3. Suggested revision to establish a better historical and cultural contextualisation of the study for readers:

- The authors speak to readers assuming unrealistic knowledge of traditional foraging as it has undergone changes in this remote Indigenous community setting. This issue may depend on how broadly the journal wishes to profile this kind of study. I am assuming that is one of the goals. If so, more specific wording will help us understand:

The extent to which the bush tomato has remained a significant part of diet, whether it has become more of a recreational food in light of dependence on purchased store foods and if so what has been the trajectory of this change. Are there data available about changing diets that can be cited more specifically? This issue is relevant to establishing the ongoing reliability of ethnographic observations over time (seemingly the last 20 years) in the bush and residential community settings. What has broad ethnographic research in this area told us about cultural change impacting hunting/foraging practices and beliefs?

Good point. We have addressed this of post-Contact mobilities in several other publications and refer to them carefully in this revised manuscript.

What have been the impacts of motor vehicle usage on such cultural changes? Some topography would no longer be accessible compared to bush dwelling practices when desert life was still operating? So are the ethnographic observations telling us about what was once done in consuming the plant species or what are the practices now?

We have a couple of papers that look at the use of motor vehicles the most recent being

Zeanah, D.W., Codding, B., Bird, D.W., Bliege Bird, R., and P. Veth 2015 Diesel and damper: Changes in seed use and mobility patterns following contact amongst the Martu of Western Australia. *Journal of Anthropological Archaeology* 39: 51-62.

We are concerned with both past and present landscape use, so have derived proxies for the way people use landscapes now and how they used landscapes prior to the 1960s.

Dreaming stories are said to have certain environmental correlates, e.g. about what are preferred food sources for emu and bustard. Can the reader learn how or why such stories might indicate long observed behaviours of these species by Martu people? This may appear obvious for anthropologists committed to awareness of both clear and subtle cultural continuities. But sociocultural research in anthropology demonstrates that religious knowledge can derive as much from spiritual convictions as from everyday interactions with bush species. Is there an assumption about the significance of Dreaming stories on the part of the authors that can be clarified more specifically? Including the authors' approach to changes over time associated with sedentisation in residential communities where religious life has gone in new directions compared with bush living in earlier generations.

Dreaming stories are mentioned explicitly to give historic context. A detailed social anthropological critique of Dreamings, origin narratives and return to homelands is beyond the scope of this paper. We also note that our Martu coauthor does not wish these secondary details to be explored in our current paper; they are considered to fall in the sacred/restricted domain.

It would also be useful for readers to learn whether interviews about cultural practices and beliefs in earlier generations have been recorded, transcribed and analysed with cross-checking across different communities and among knowledgeable individuals. This to control for idiosyncratic versions of beliefs about species and spiritual entities not being adopted uncritically.

Beliefs in earlier generations have not been recorded; recalling these are the last societies on the planet to make contact with settler societies. This request is beyond the scope of the paper.

Some or all of the authors have completed ethnographic research with Martu people over the past 20 years. The reader would benefit from a succinct clarification as to how cultural changes in foraging behaviours have been addressed generally. What has been the role, if at all, of strategic traditionalism that may have resulted in self-conscious fashioning in usage of bush resources? Is the bush tomato a high-status bush food more so or less so now than in the past? As it stands there are a few snippets revealing knowledge the investigators may have of cultural changes, such as lines 327-8 noting the small number of elders with actual bush dwelling knowledge about camp sites and presumably associated foraging practices. The authors, though presumably not all of them, have carried out long-term community-based ethnographic research, and the reader would benefit hearing of the scope of those studies that provide the context for this particular paper.

We added some more text to address this issue, however such data are not critical to the hypotheses under test. The suggestion might represent a productive paper in the future with a senior Western Desert anthropologist.

- I suggest citing recent debates about absence / presence of agriculture in Australian Indigenous pre-colonisation modes of subsistence. Indeed, I don't see how the paper can be published without at least acknowledging this debate. Given the importance of the debate for contemporary Australian society there should be at least a note stating or acknowledging the authors' position on the works of Pascoe, Sutton & Walshe and other scholars where relevant. My reading of the article is that the authors take as a starting point a lack of what is normally regarded scientifically as agriculture across arid Australia in pre-European settlement times. However, readers will benefit from an explicit comment on this issue.

We have summarised the core arguments of the Dark Emu debate as these are relevant to seed dispersal and harvesting. We have cited the key authorities should readers wish to pursue the topic.

Reviewer #3 (Remarks to the Author):

This work will be of significance to people in the field. There are many studies in archaeology/archaeobot/ethnobot where researchers have made observations about the co-occurrence of specific plants and sites of human occupation. However, I cannot think of any that have made attempts to provide quantitative data of this distribution along with qualitative observations and to provide control samples in doing so. For that reason alone, this is a paper that provides important new data for researchers. This work also makes use of data from a long-term study of a region of the Western Desert (Martu peoples) that further shows the value of such long-term research relationships.

I think that the paper is well produced with good supporting data and good quality illustrations and figures. The methods appear sound, showing positive or negative correlations in the presence/absence of key species against a combination of anthropogenic and non-anthropogenic variables. Throughout, there is an attempt to isolate variables or combinations of variables that are more strongly linked to human activity.

I have a few minor observations. Some of these may arise from my lack of familiarity in the statistical methods employed and the manner of data presentation. I am not familiar with the software employed for this study.

Spelling: line 422 of 'fiire'

This has been rectified.

On page 9 authors refer to five anthropogenic predictors, but then do not list them as they do 9 (nine?) most informative, non-collinear.... I could identify four: distance to

ethnohistoric site, water sources, site type, and winter fires. Perhaps list the anthropogenic predictors after a colon?

This issue should now be clarified in the text under hypotheses and predictions

Water permanence. Line 317 states use of a permanence scale from 1 to 4, but these are summed in the statistics to 'water permanence'? Do the values of 1 to 4 have any role in the data presented?

We have checked the text and are unclear about what this comment is referring to. The new tables and captions may have addressed this perceived issue.

Figure 1, line 444. Location of observed dispersal sites (brown). In my copy, I think these sites are marked with a red open circle with a red open cross.

A new map has been generated to correct this issue.

Figure 2. combined variables on the Y-axis shown as fire density x water perm and site dist x fire density. What does the 'x' refer to? Are values being multiplied? In Figure 3. combination variables separated by a colon, e.g. dis site: Win fire. Is this a ratio variable? Needs clarification.

This has been clarified in the text. X refers to interaction terms in the multivariate analysis (crossed terms).

Figure 3. Non-specialists might appreciate a bit more text somewhere on the use of odds ratios and what is being displayed/shown with this data.

we now define the concept of odds ratios in the captions.

Table 1. Under column prediction, the term 'effect' is used as either positive or negative - but effect in what sense? As correlation? i.e. there is a positive correlation between site distance and presence (or frequency/ha?) of Solanum d? For scaevola, it is noted that model outcome Anth+fire for site distance is 'farther from sites'. But what is meant here - is this qualitative or quantitative? Some explanatory text could be included with this table to help the reader navigate this summary data presentation.

Effect size is a statistical term that refers to the model coefficient. A model coefficient reports the increase in the independent variable with a one unit change in the dependent variable, and in binomial logit regression, are reported as exponentiated odds ratios. In our case, the variables were standardized as is standard practice to facilitate the comparison of effect size between different predictors. These explanations are now given in the figure/table captions.

Reviewers' Comments:

Reviewer #1:

Remarks to the Author:

The authors have addressed all the comments in the original review. The revised manuscript reads well, the methods are easy to understand. I would recommend for publication.

Reviewer #2:

Remarks to the Author:

1. Yes, overall, my points have been addressed in the revisions.

2. A small point is that i suggest inserting a citation for the relevant Native Title determination documentation that is publicly available. This to assist readers wishing to follow up the context of this important aspect of Martu land connections and living circumstances. Native title rights enable and facilitate the kind of continued use of bush resources documented by the authors. Certainly, this will be so into the future for younger generations.

3. A second small point is i suggest replacing the term 'today', which i think is used several times to indicate presumably 2024, or the time of writing, or perhaps the period over which the research has included fieldwork among Martu people. Readers in future years will need to know what 'today' refers to.

Reviewer #3:

Remarks to the Author:

The authors have responded positively to the recommendations and requests for clarification from all three reviewers. Additional information and correction has clarified the paper for publication.

This is an important paper and I look forward to publication.

A minor note: I could not find mention of (Figure 2) in the main text. It may have been omitted in edits?

In caption text of Figure 2 there is a cross-reference to Table 3 but no cross-reference to Figure 2 in the Table 3 text.

Why is the word 'matters' underlined on pg 9 line 425?

Point by point response

Reviewer #2:

2. A small point is that i suggest inserting a citation for the relevant Native Title determination documentation

Done.

3. A second small point is i suggest replacing the term 'today', which i think is used several times to indicate presumably 2024, or the time of writing, or perhaps the period over which the research has included fieldwork among Martu people. Readers in future years will need to know what 'today' refers to.

We deleted or clarified all instances of 'today'.

Reviewer #3:

A minor note: I could not find mention of (Figure 2) in the main text. It may have been omitted in edits?

Yes it was omitted, we added it back.

In caption text of Figure 2 there is a cross-reference to Table 3 but no cross-reference to Figure 2 in the Table 3 text.

Added the cross reference.

Why is the word 'matters' underlined on pg 9 line 425?

Deleted the underlining as it wasn't allowed anyway.